# Cancer patient survival can be parametrized to improve trial precision and reveal time-dependent therapeutic effects

Deborah Plana [1,2], Geoffrey Fell [3], Brian M. Alexander[3,4], Adam C. Palmer [5,6✉] & Peter K. Sorger [1,6✉]

Individual participant data (IPD) from oncology clinical trials is invaluable for identifying factors that influence trial success and failure, improving trial design and interpretation, and comparing pre-clinical studies to clinical outcomes. However, the IPD used to generate published survival curves are not generally publicly available. We impute survival IPD from ~500 arms of Phase 3 oncology trials (representing ~220,000 events) and find that they are well fit by a two-parameter Weibull distribution. Use of Weibull functions with overall survival significantly increases the precision of small arms typical of early phase trials: analysis of a 50-patient trial arm using parametric forms is as precise as traditional, non-parametric analysis of a 90-patient arm. We also show that frequent deviations from the Cox proportional hazards assumption, particularly in trials of immune checkpoint inhibitors, arise from time-dependent therapeutic effects. Trial duration therefore has an underappreciated impact on the likelihood of success.

[1] Laboratory of Systems Pharmacology and the Department of Systems Biology, Harvard Medical School, Boston, MA, USA. [2] Harvard-MIT Division of Health Sciences and Technology, Harvard Medical School and MIT, Cambridge, MA, USA. [3] Dana-Farber Cancer Institute, Boston, MA, USA. [4] Foundation Medicine Inc., Cambridge, MA, USA. [5] Department of Pharmacology, Computational Medicine Program, UNC Lineberger Comprehensive Cancer Center, University of North Carolina at Chapel Hill, Chapel Hill, NC, USA. [6] These authors contributed equally: Adam C. Palmer and Peter K. Sorger. ✉email: palmer@unc.edu; peter_sorger@hms.harvard.edu

Extensive effort has been devoted to increasing rates of success in oncology drug development by improving preclinical studies[1–3]. However, completed randomized controlled trials (RCTs) remain the most valuable single source of information for understanding opportunities and challenges in drug development. Retrospective comparison of trials is most commonly performed via meta-analyses and systematic reviews[4] with the goal of improving patient management in specific disease areas[5]. Retrospective analysis has also been credited with improving the statistical treatment of trial data, which can be complex and confounded[6]. However, quantitative analysis of oncology trials is difficult to perform at scale because individual participant data (namely times of progression, death, and censoring; IPD), which are necessary for high-quality analysis, are rarely available[7,8]. Trial results are instead reported in the form of summary statistics and, in the case of oncology trials, plots of patient survival based on the Kaplan–Meier estimator[9]. These plots are generated using IPD but it has proven time-consuming and resource-intensive to gain access to the underlying IPD values because journals and investigators do not generally make them available[10].

To address this problem, the International Committee of Medical Journal Editors (ICMJE) recently developed a set of data reporting standards to encourage the release of IPD from all clinical trials[11]. However, less than 1% of papers published in a two-year period were found to have made IPD publicly available[12]. We and others have developed methods to bypass this problem by using image processing to impute IPD values from published plots of the Kaplan–Meier estimator[13–15]. In this manuscript, we describe a comprehensive analysis of imputed IPD and reconstructed survival curves from ~150 publications reporting Phase 3 cancer trial results, which in aggregate comprise ~220,000 overall survival or event-free survival events (e.g., progression-free survival, PFS). We also make these data freely available via an interactive website (https://cancertrials.io/) and through the NCI-recognized Sage Synapse repository (ID: syn25813713)[16]. Our approach is consistent with the Institute of Medicine's reports on best practices for sharing data from published clinical trials, including crediting the sources of the data and sharing all code used in the analyses[17].

Analyzing survival functions with parametric forms of different types has a long history[18], but evidence has been lacking about which distribution best represents real data. Parametric statistics are also well known to increase precision, but only when the fit to data is sufficiently accurate. We now show that therapeutic responses for multiple cancer types and therapeutic classes as measured both by overall survival (OS) and event-free survival (e.g., progression-free survival; PFS) are well fit by unimodal distributions described by the two-parameter Weibull function; one parameter is proportional to median survival and the second quantifies changes in hazard over time. Using Weibull functions, we find that a 50-patient trial arm (assessing overall survival) is as accurate and precise as a 90-person arm evaluated using traditional nonparametric statistics; this finding is directly applicable to improving the precision of therapeutic efficacy estimates made with the small patient populations typical of Phase 1 and 2 oncology trials[19]. Weibull fitting of survival data also confirms that violations of the assumption of proportional hazards are common in contemporary Phase 3 trials[15] notably for immune checkpoint inhibitors (ICIs)[13,20]—but also more broadly. Violations arise from variation in hazard ratios over time and, as a consequence, so does the likelihood of trial success (which is most commonly defined as a hazard ratio less than one at a 95% confidence level). This effect is different from the increase in statistical confidence that occurs in any trial as a result of the accrual of more events. In particular, simulation suggests that some failed trials with strong time-dependence might have been judged to be successful had they been run for slightly longer. Trial characteristics computed from IPD allow for comparison of response distributions across diseases and therapeutic modalities, potentially making it possible to improve the design of future trials and reduce attrition. The accuracy of the Weibull form in describing survival data may also assist cost-effectiveness research in which diverse parametric statistics are already in use[21].

## Results

**Cancer patient survival can be accurately parameterized**. We used previously described algorithms and approaches[14,15] to mine published papers reporting the results of Phase 3 clinical trials in breast, colorectal, lung, and prostate cancer with endpoints including OS or surrogates such as PFS, disease-free survival (DFS), and locoregional recurrence (LRR) (which we henceforth consider in aggregate as "event-free survival"). For each trial between 2014 and 2016 that met our search criteria, plots of the Kaplan Meier (KM) estimator were extracted from trial figures using the DigitizeIt software (version 2.5.3; Braunschweig, Germany), while the at-risk tables and the number of patient events were manually extracted from the publication. We then used the digitized KM survival curves to estimate patient-level time-to-event outcomes (IPD; e.g.,: times of progression, death, and censoring; Fig. 1a). We recently reported the use of this approach to reconstruct patient-level data for oncology Phase 3 clinical trial publications identified through a PubMed search[22]. Study-level information such as cancer type, metastatic status, treatment modality, and trial success was also manually curated.

Analysis of OS data from 116 published figures yielded 237 distributions (91,255 patient events). Data on event-free survival from 146 figures yielded 301 distributions (127,832 patient events). Classes of therapy included chemotherapy, ICIs, radiotherapy, surgery, targeted therapy, and placebo/observation. All imputed data were compared against the original trial publication for accuracy[22] and trials with inaccuracies in the imputation procedure were excluded. The accuracy of IPD imputation is discussed further in Supplementary Data 1 and "Methods". The data set is released in its entirety as Supplementary Materials to this paper, via an interactive website (https://cancertrials.io/), and through the online Synapse repository (ID: syn25813713)[16].

A variety of parametric forms have been proposed to describe survival in oncology trial data, including the Log-Normal, Log Logistic, Gamma, Weibull, Gompertz–Makeham, and Exponential distributions[23,24]. These differ in their hazard functions, which quantify the likelihood of an event (e.g., death or progression) at a given time[25–28]. To our knowledge, no systematic assessment of the accuracy of these forms in describing empirical data from a large set of oncology trials has previously been described. We therefore assessed the goodness-of-fit ($R^2$) of different parametric distributions to imputed IPD. First, "best-fit" parameter values were estimated for individual IPD distributions using a maximum-likelihood procedure (Fig. 1b, c). Second, mathematical transformations specific to a parametric form were used to linearize the distribution of event times and the corresponding survival values (in the case of the Weibull form the linearization is the Weibull plot)[29]. Data that perfectly follow a proposed distribution would, in the transformed form, follow a straight line with $R^2 = 1$.

Parametric forms for survival distributions differ the most at long follow-up times when the tails of the distributions fall to an asymptotic value or to zero. However, such long event times are rarely recorded in traditional oncology trials, which are limited in duration by cost and increased censoring (often because patients switch to an alternative therapy). As a consequence, we found

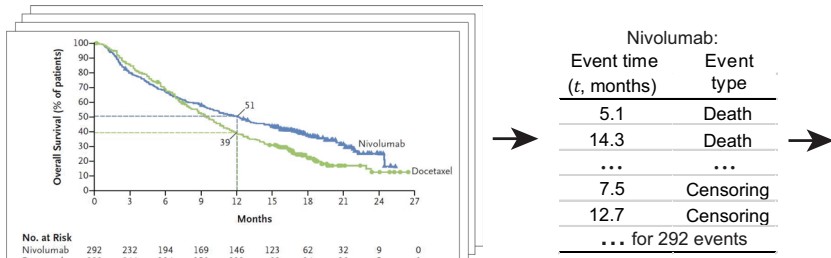

**a** Reconstruct event tables from published Kaplan-Meier estimators and at-risk tables

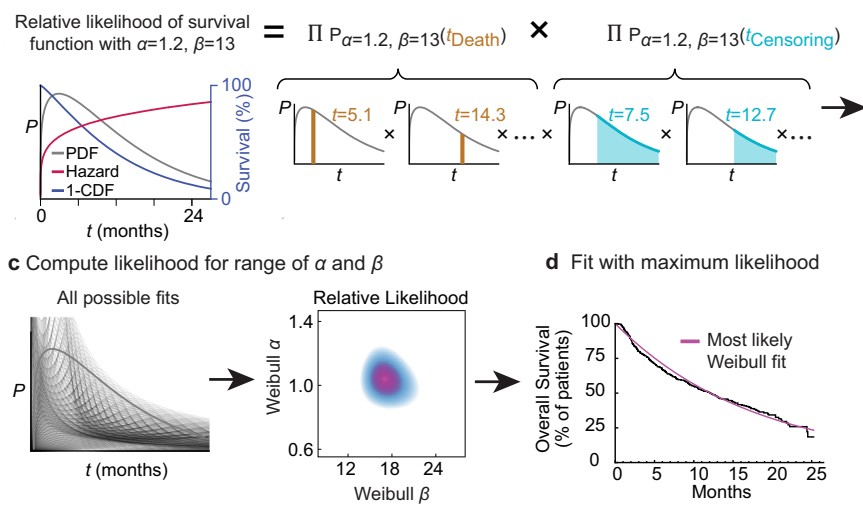

**b** Calculate likelihood of observing patient data given a set of Weibull parameters

**c** Compute likelihood for range of α and β

**d** Fit with maximum likelihood

**Fig. 1 Procedure for parameterizing survival curves starting with published figures. a** Kaplan–Meier survival curve and at-risk table obtained from clinical trial publication. Individual participant data (IPD) were imputed from digitized survival curves and at-risk tables as previously described (see "Methods"). **b** Each set of parameters corresponds to a different probability density function (PDF) and survival function (which corresponds to 1 minus the cumulative density function (CDF)). The likelihood of observing actual data is then computed. **c** Likelihood calculation is repeated for a set of possible parameter values. **d** The most likely (best) fit is obtained by finding the parameter values with the maximum likelihood.

that two types of two-parameter distributions fit survival data equally well: Weibull distributions and Log-Normal distributions (Weibull median $R^2 = 0.981$ and Log-Normal median $R^2 = 0.980$). We chose to use the two-parameter Weibull distribution because its parameters are easily interpreted in terms familiar to oncologists. The Weibull α (shape) parameter describes increasing or decreasing hazard over time[30], and the β (scaling) parameter is proportional to median survival time[31]. Survival data fit by Weibull distributions having $α < 1$ have decreasing hazard rates over time, meaning that the likelihood of progression or death is highest at the start of the trial and then falls. A value of $α = 1$ corresponds to a constant hazard and $α > 1$ to a hazard that increases with time.

For each trial arm in our data set, we obtained best-fit values for α and β (Fig. 1d). The distributions for individual arms and their parameterizations can be visualized in three different ways: (i) as probability density functions (PDFs), the likelihood that an event will occur at any particular time $t$; (ii) as cumulative density functions (CDFs), the integral of the PDF with respect to $t$; for OS data, 1-CDF is overall survival at $t$; and (iii) as hazard functions, which correspond to the ratio of the PDF and survival function. In oncology trials, the survival function is usually determined using the nonparametric Kaplan–Meier estimator (Fig. 2a–c), which accounts for progression or death events as well as censoring (e.g., withdrawal of a participant from the trial, or loss of follow-up, for reasons other than progression or death). A plot of patient-level data as a PDF shows that death or progression is

right-skewed for all values of α that we observed in trial data (as illustrated in Fig. 1b). Thus, a substantial proportion of all events occur well after the modal (peak) survival value. Fitting Weibull distributions, therefore, quantifies the frequently observed phenomenon that the response of a subset of patients to therapy is substantially better than the most commonly observed response to that treatment.

For trials reporting OS data, we found that a two-parameter Weibull distribution had a median coefficient of determination of $R^2 = 0.981$ (lower quartile, Q1: 0.966, mean: 0.975, upper quartile, Q3: 0.989) across 237 trial arms from 116 figures in clinical trial reports (Fig. 2d; "Methods"); the histogram of $R^2$ values for every OS arm of every clinical trial can be found in Fig. 2e. The theoretical maximum $R^2$ value can be calculated under the hypothesis that all OS distributions are Weibull distributions and that deviations are attributable only to sample size variability, which yields a maximum $R^2 = 0.995$. Thus, ~2% of variance observed is not explained by the Weibull model. For trials reporting event-free survival data (e.g., PFS; Supplementary Fig. 1) median $R^2 = 0.950$ (Q1: 0.909, mean: 0.929, Q3: 0.970) as compared to a theoretical maximal $R^2 = 0.996$, which corresponds to 5% of observed variance not explained by the Weibull model. Biomarker-stratified arms were also well described by a single two-parameter Weibull distribution. This is illustrated in Supplementary Fig. 2a, b for Weibull fits to OS and PFS data for panitumumab in combination with FOLFIRI and for FOLFIRI alone in wild-type and mutant *KRAS* metastatic

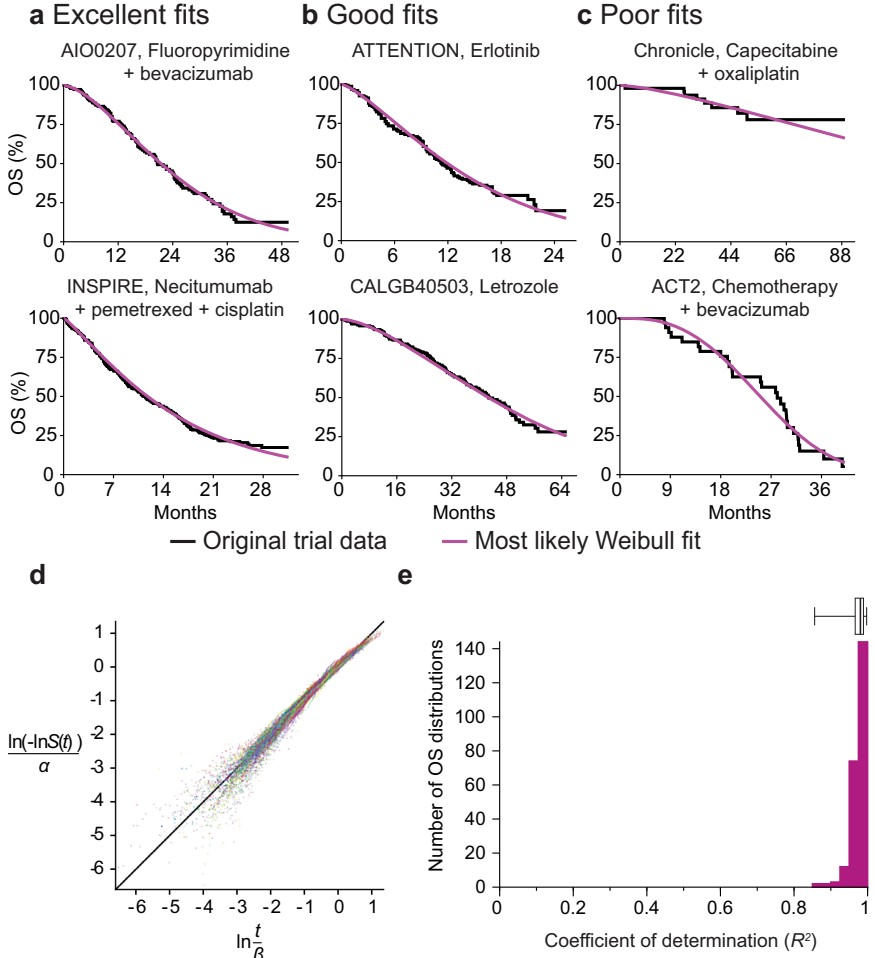

**Fig. 2 Representative fits of Weibull distributions to overall survival (OS) trial data. a** Weibull fits to data for plots of the Kaplan–Meier estimator falling in the top 25th percentile quality of all fits (NCT00973609, NCT00982111) **b** between the first and third quartile (NCT01377376, NCT00601900) and **c** in the lower quartile (NCT00427713, NCT01229813). **d** Weibull plot for patient events in all OS trials (for 237 trial arms from 116 publication figures). This is a transformation of survival data that is linear if the data follow a Weibull distribution. **e** Goodness of fit as the coefficient of determination ($R^2$) explained by fitted Weibull functions for all trial arms reporting OS data ($n = 237$ trial arms). Boxplot center line indicates median; box limits indicate the upper and lower quartile values; whiskers span the full data set.

colorectal cancer (trial 20050181[32]; average $R^2 = 0.99$ for OS curves and 0.92 for PFS curves). We conclude that a two-parameter Weibull distribution provides an excellent fit, with the few exceptions discussed in detail below, to available trial data across multiple types of cancer, treatment modalities, and metastatic status.

**Investigating the least good fits of survival data to Weibull parametrizations**. Across the entire data set, some of the worst fits for two-parameter Weibull forms were observed for trials with relatively few events, for example, the Chronicle trial (NCT00427713[33]) with only eight deaths in the treatment arm and 16 progression events in the observation arm (Fig. 2c; Supplementary Fig. 1). Fit was also poor for trials involving pre-planned changes in treatment such as the ACT2 trial (NCT01229813[34]), in which treatment induction was followed by randomization to maintenance treatment at 18 weeks (Fig. 2c). In cases such as this, responses varied over the course of the trial by design, and a good fit to a single two-parameter survival function is not expected.

For a small number of trials in which the asymptote of the fitted survival curve was greater than zero (i.e., patients were expected to be alive at the end of the longest follow-up), a three-

parameter Weibull distribution consisting of the traditional two-parameter distribution with an additional "cure rate" term[35] had an improved fit (Supplementary Fig. 3). Since the improvement in fit was modest and two-parameter Weibull forms are both more computationally tractable and are more parsimonious, we relied on them for all subsequent large-scale analysis. However, the use of a cure rate parameter might nonetheless be advisable for different sets of data in which cure is a known outcome (e.g., R-CHOP for non-Hodgkin's lymphoma[36]).

We also observed that event-free survival exhibited a slightly poorer fit to Weibull forms than OS data (5% vs 2% of observed variance not explained). Inspection of the relevant curves showed that this was caused primarily by a sharp decrease in survival at early time points and a shallowed decrease subsequently; this behavior has previously been interpreted as evidence for subpopulations of responding and non-responding patients, particularly in trials of ICIs[37]. It has also been attributed to delayed T-cell activation by ICI therapy[38]. For trials of these agents, we found that fits to PFS data could be improved by using a mixture model comprising two different two-parameter Weibull distributions, each with its own $\alpha$ and $\beta$ parameters. This is potentially consistent with a two-population hypothesis (Fig. 3a; Supplementary Fig. 4). However, a mixture model also resulted in

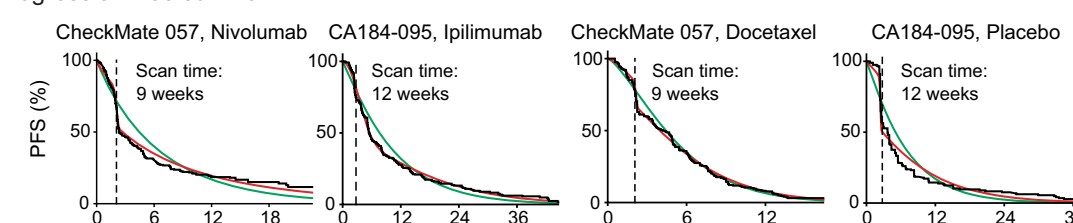

**a** Progression-free survival

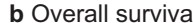

**b** Overall survival

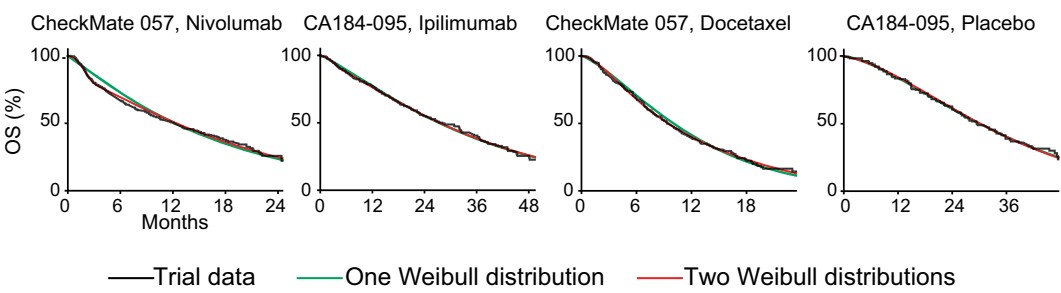

—— Trial data —— One Weibull distribution —— Two Weibull distributions

**c** Weibull fit where progression is observed at scan times

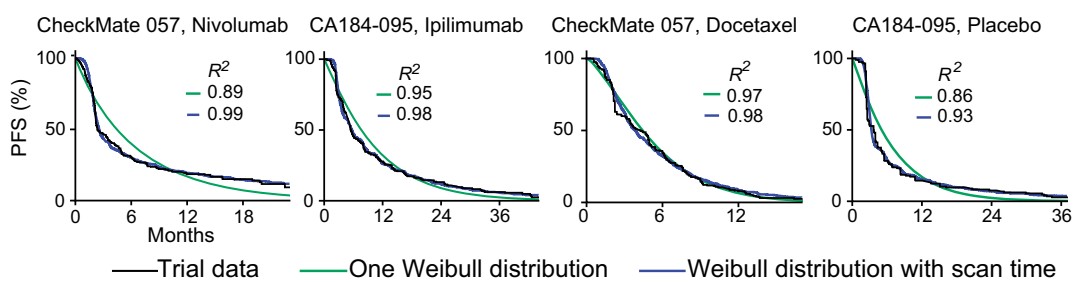

—— Trial data —— One Weibull distribution —— Weibull distribution with scan time

**Fig. 3 Fit of Weibull models to progression-free survival and overall survival data for trials of immune checkpoint inhibitors. a** PFS distributions and **b** OS distributions for individual arms of two trials of immune checkpoint inhibitors (NCT01673867, NCT01057810) with fit to either one or two Weibull distributions. PFS data are best described by using a mixture model of two Weibull distributions, each with two parameters. OS distributions are described equivalently well by a one or two-distribution fit. **c** PFS simulations that account for the periodicity of radiological scans to detect progression improve the quality of one-distribution Weibull fits, as quantified by the coefficient of determination ($R^2$).

a better fit to the control arms in these trials, suggesting that the deviation from a single distribution was not ICI-specific. Moreover, a mixture model exhibited no meaningful improvement in fit as compared to a single-distribution for OS data from ICI or control arms in these trials (Fig. 3b; Supplementary Fig. 4). Finally, when we examined PFS data from an additional 25 ICI trials, we found that the drop in survival at early event times (which we identified as the time $t$ corresponding to the greatest change in the slope of the survival curve) was strongly correlated with the time of the first radiological scan (as reported in trials' methods sections; Pearson correlation 0.982, $p < 10^{-21}$; Supplementary Data 3). We surmised that fitting Weibull distributions to PFS data was confounded by scan times. To test this idea we simulated the influence of scan times by taking a single two-parameter Weibull distribution and imposing a scanning interval of nine or twelve weeks (the actual value was extracted from the trial protocol). This generated the steep decline in PFS values observable in the control and experimental arms of actual ICI trials, and improved fit to PFS distributions, raising mean $R^2$ from 0.93 to 0.98 (Fig. 3c; Supplementary Fig. 4; "Methods"). We conclude that a steep drop in initial PFS is likely to arise because values at early time points from a unimodal response distribution are concentrated in time by scans performed at discrete intervals. We further conclude that mixture models involving two Weibull curves are not necessary to accurately describe survival for ICIs or

any other class of therapy that we have examined. Instead, when scan times are accounted for, single two-parameter Weibull distributions are found to have an excellent fit to PFS data ($R^2 = 0.98$).

**Parametric fitting improves the precision of drug efficacy estimates**. To compare the performance of Weibull-based and nonparametric methods used for survival analysis we calculated pointwise confidence intervals (at 12-months). This is a frequently reported statistic for many early phase oncology trials (Phase 1 and 2) that involve relatively small numbers of patients. It is also a landmark outcome in systematic reviews and meta-analysis of oncology clinical trials[39], and used to guide the design of larger trials. A challenge in the analysis of such data is that, when too few events have occurred, nonparametric numeric confidence estimates return non-informative values (usually reported as a value "not reached" or "indeterminate"[27] as illustrated for two different scenarios in Supplementary Fig. 5a). To determine how parametric analysis would perform in this setting, we subsampled groups of 20–100 patients at random from the arms of imputed Phase 3 trials. We then compared estimates of 12-month survival for small cohorts with a well-powered ground truth value obtained using the full Phase 3 data set. We found that 12-month survival estimates were non-informative for 20–40% of OS trial arms, and 23–61% of event-free survival trial arms, with

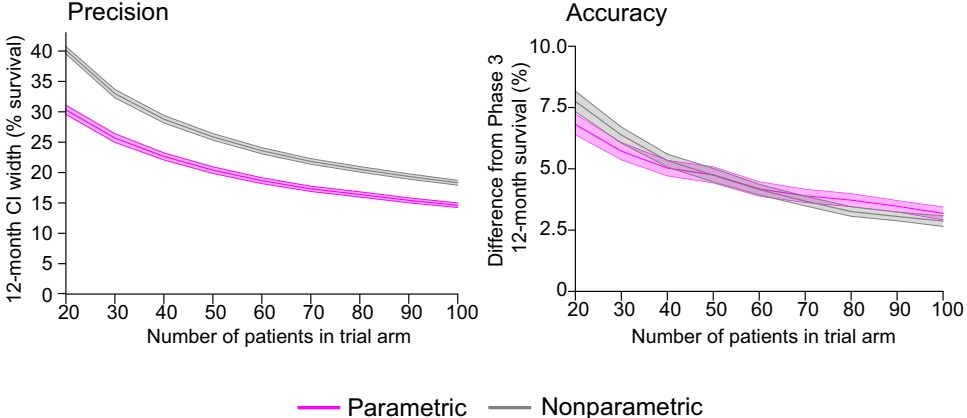

**Fig. 4 Impact of parametric fitting on the precision and accuracy of overall survival estimates.** Comparison of nonparametric and parametric (Weibull distribution) methods to compute 12-month OS confidence intervals (CIs) for trials with small cohorts (20–100 patients) produced by randomly subsampling patient events from 125 actual trial arms. Note that nonparametric estimates did not return an informative confidence interval for 41% of OS curves (out of 213 OS curves with at least 100 patients), while Weibull fitting made it possible to calculate 12-month confidence intervals for every survival curve in every simulation. Precision is defined as the width of the confidence interval in percent survival. Accuracy is defined as the absolute difference between the 12-month survival estimated from small cohorts and the value computed from all patients in a given Phase 3 trial arm. Lines denote mean values and shaded regions are 95% confidence intervals.

higher failure rates occurring when the sample size was smaller (Supplementary Fig. 5b, c). In comparison, the use of a Weibull parametric form made it possible to calculate 12-month confidence intervals for all of the ~45,000 simulated 20–100 patient trials that we examined.

Considering only the subset of OS curves for which confidence intervals could be computed using both parametric and nonparametric methods (125 curves), we found that a Weibull-based approach was more precise (it had a narrower confidence interval) across all sample sizes, and accuracy was comparable. By way of illustration, the precision of a 50-patient trial arm was comparable to that of a 90-patient study using traditional methods (Fig. 4). For event-free survival (for which 99 event-free survival curves could be compared), the precision of a Weibull-based approach was also greater than a nonparametric approach across all sample sizes, while accuracy was equivalent for small sample sizes (fewer than 40 patients) (Supplementary Fig. 6a). We conclude that the use of Weibull distributions to parameterize data from small trials reporting either OS or PFS data approximately doubles the number of trials for which informative confidence intervals can be determined for a point survival estimate (e.g., survival at 12 months). Moreover, for the subset of trials in which parametric and nonparametric methods can be directly compared, the former is as precise using roughly half the number of patients.

Figure 4 indicates that survival estimates made using Weibull parameterization decrease in relative accuracy as compared to nonparametric methods as patient number is increased. This arises simply because nonparametric analysis of the full set of Phase 3 data was defined as the ground truth. It is nonetheless true that using Weibull forms is most valuable when cohorts are small (fewer than ~40 patients). This sample size is typical of Phase 1 or 2 oncology trials, a setting in which alternative statistical methods are also most likely to be acceptable from a regulatory standpoint.

As one illustration of the use of Weibull parameterization, we analyzed a recent trial that encompassed both Phase 1 and 2 data and tested pembrolizumab with dabrafenib and trametinib for metastatic *BRAF*-mutant melanoma (MK-3475-022/KEYNOTE-022; NCT02130466)[40]. Parametric fitting for 15 patients in Phase 1 yielded a median PFS of 14.8 months and 95% confidence interval of 7.8–23 months, while nonparametric estimates yielded a median value of 15.4 months and 95% confidence interval of 5.4 months to "not reached." Nonparametric analysis of a Phase 2 cohort of 60 patients for this same trial revealed a median PFS of 16 months and a 95% confidence interval of 8.6–21.5 (ref. [41]). Thus, parametric fitting of data from 15 patients made a comparably precise and accurate estimate of median PFS as nonparametric analysis of 60 patients (Supplementary Fig. 6b). The availability of a more precise parametric approach would in principle have made it possible to use the same number of patients enrolled in this Phase 2 study (e.g.,: 60 patients) to perform three different signal-finding studies (each involving 20 patients) with no loss of statistical power. This would have been particularly helpful in the case of KEYNOTE022, a trial that failed to meet its primary endpoint.

**Evaluating the impact of patient heterogeneity on the accuracy of Weibull parameterization.** Subsampling Phase 3 trials to generate synthetic arms having the small numbers of patients typical of Phase 1 and 2 trial cohorts has the advantage that the Phase 3 data serve as the ground truth. However, it has the disadvantage that patient populations in early stage trials are often more heterogenous than in pivotal trials. We have been unable to identify a sufficient number of matched early and late phase survival data for comprehensive investigation of this issue. As an alternative approach, we simulated a trial having a heterogeneous population of ~900 patients with a mixture of breast, colorectal, lung, and prostate cancer cases. The simulated cohort was constructed by subsampling five patients from each of 172 trial arms that reported OS for patients with metastatic cancer. We observed that for a representative simulation, a two-parameter Weibull form accurately described the synthetic trial data ($R^2 = 0.98$; Supplementary Fig. 7). However, a representative synthetic cohort involving patients drawn from both metastatic and local cancers (237 trial arms) is not as well fit by a two-parameter model ($R^2 = 0.95$) but fit improves with the addition of a cure-rate parameter ($R^2 = 0.98$). From this analysis, we conclude that trial arms having different types of solid tumors, as encountered in some basket trials, can be accurately parameterized by two-parameter Weibull functions. If metastatic and non-metastatic

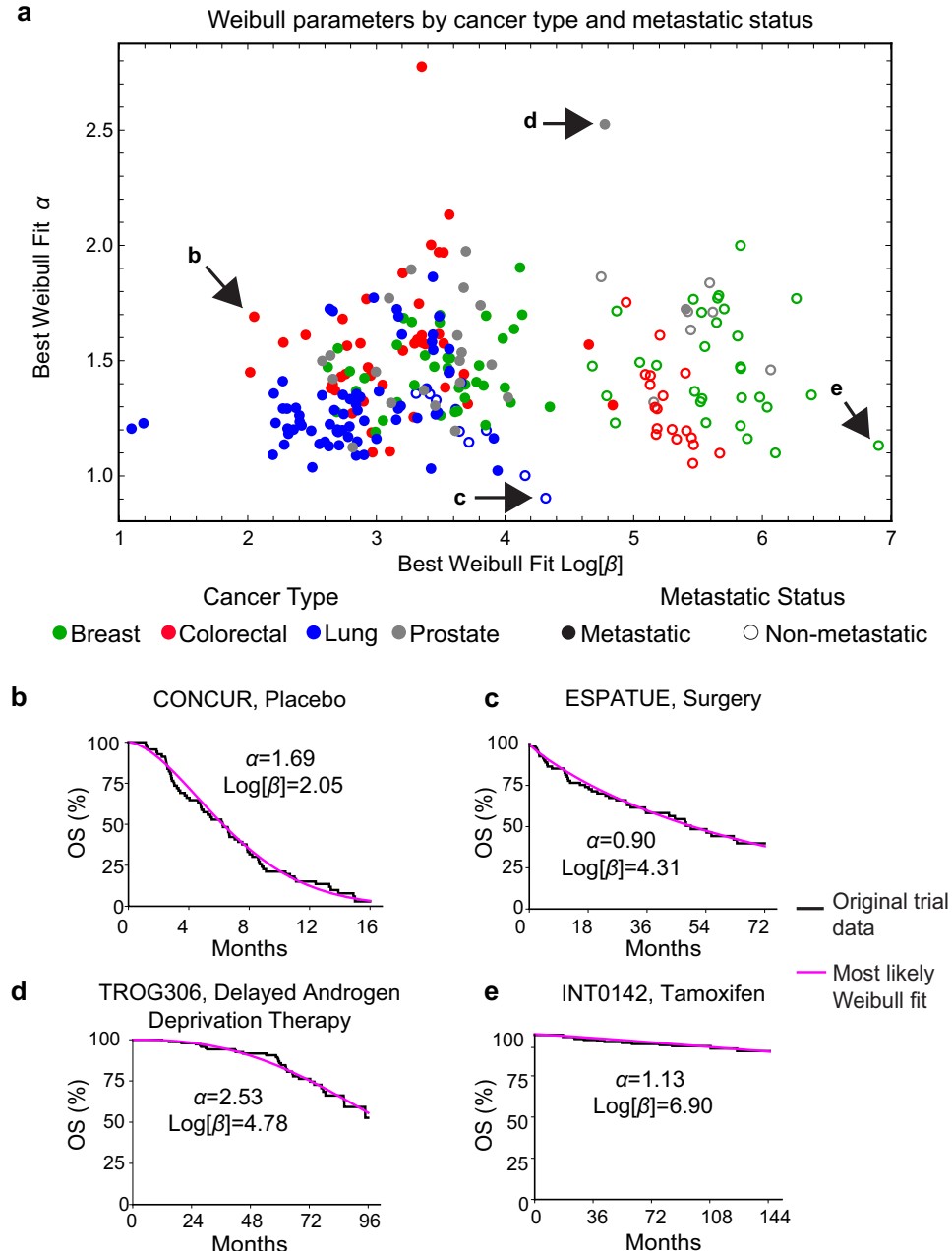

**Fig. 5 Best-fit Weibull parameter values for trials reporting overall survival (OS) data.** Weibull fits for trials reporting OS data, encompassing 237 survival curves from 116 trial figures. **a** Survival distributions categorized by cancer type and metastatic status (defined as trials that included patients with distant metastases). Representative survival functions and fits for trials across a variety of cancer types including **b** metastatic colorectal cancer (NCT01584830) **c** non-metastatic lung cancer (NCT number not reported) **d** metastatic prostate cancer (NCT00110162) and **e** non-metastatic breast cancer (NCT number not reported).

disease are mixed in an RCT, the addition of a third parameter is likely to improve fit.

**Weibull fitting quantifies survival differences across cancer types**. The availability of a large set of IPD made it possible to search for systematic differences in the parameters of survival distributions by disease class. Best-fit Weibull parameters were compared across cancer types and metastatic status using an ANOVA test with a Bonferroni correction for multiple hypothesis testing at a two-tailed significance level of 0.05 (see Supplementary Data 2). The largest difference in parameter values was between metastatic and non-metastatic disease, irrespective of

tumor type ($\beta$ values corresponding to median survival of 22 and 180 months respectively). We also observed that $\beta$ values were significantly larger for breast cancer than lung cancer in the metastatic setting ($\beta$ values corresponding to median survival of 28 versus 14 months) (Fig. 5), which is consistent with previous data on relative disease severity[42,43]. Parameter values for trials reporting event-free survival followed a similar pattern to OS values (Supplementary Fig. 8 and Supplementary Data 2). Lung cancers had a significantly lower $\alpha$ (shape parameter) for OS as compared to other cancer types (average $\alpha = 1.30$ for lung; versus ~1.5 to 1.6 for breast, colorectal, and prostate cancers), demonstrating a relatively high probability of early death. This difference in shape also corresponds to a wide distribution of lung cancer

**a** Weibull Parameters for OS Trials

**Fig. 6 Parameter values for Weibull fits to overall survival (OS) data scored by trial outcome. a** Differences in Weibull $\alpha$ and $\beta$ parameters for experimental and control arm OS data (121 comparisons from 116 trial figures). For $\alpha$, the value for the control arm was subtracted from the value for the experimental arm. Differences in $\beta$ were computed by determining the percent change in $\beta$ in the experimental arm with respect to the control arm (positive values indicate larger $\beta$ in the experimental arm). Asterisks denote trials that tested immune checkpoint inhibitors (ICIs). Success in all cases was judged based on the original report and most often corresponded to a hazard ratio less than one at 95% confidence based on Cox proportional hazards regression. Published hazard ratios and associated confidence intervals (red lines) and mean Weibull ratios of cumulative hazards (blue lines) with corresponding 95% confidence intervals, along with the reconstructed survival curves, of four clinical trials in the data set: **b** NCT01377376 (tivantinib plus erlotinib vs. erlotinib), **c** NCT01607957 (TAS-102 vs. placebo), **d** NCT01673867 (nivolumab vs. docetaxel), **e** NCT00861614 (ipilimumab vs. placebo after radiotherapy).

survival times as compared to other cancer types, which may reflect heterogeneity in lung cancer trial cohorts. We propose that parameters drawn from IPD be used to model cancer survival distributions for future exploratory trials and to facilitate inter-group comparisons in master protocol or basket trials, which often involve different cancers types[44,45].

**Impact of trial length on estimates of relative hazard**. Randomized controlled trials in cancer are conventionally evaluated based on the use of Cox regression to estimate the semi-parametric hazard ratio (hereafter referred to as $HR_{SP}$). If the hazard ratio is significantly below one then the test treatment decreases the risk of death or progression relative to control, and the trial is regarded as successful[27,46]. As expected, when Weibull

$\alpha$ and $\beta$ parameter values were compared between experimental and control arms, a trial was more likely to be successful (which, following common practice, we defined as $HR_{SP} < 1$ at a 95% confidence level) when differences in $\beta$ values were larger: the median difference between control and experimental $\beta$ values in OS curves was $-0.6\%$ for unsuccessful trials and 29% for successful trials (Fig. 6a). A similar pattern was observed for event-free survival data, with control and experimental $\beta$ values differing by 1.0% for unsuccessful trials and 36% for successful trials (Supplementary Fig. 9).

Fundamental to the model of proportional hazards is the idea that the hazard functions for control and experimental arms are related by a constant of proportionality (the hazard ratio) that does not change over time. However, prior work has shown that

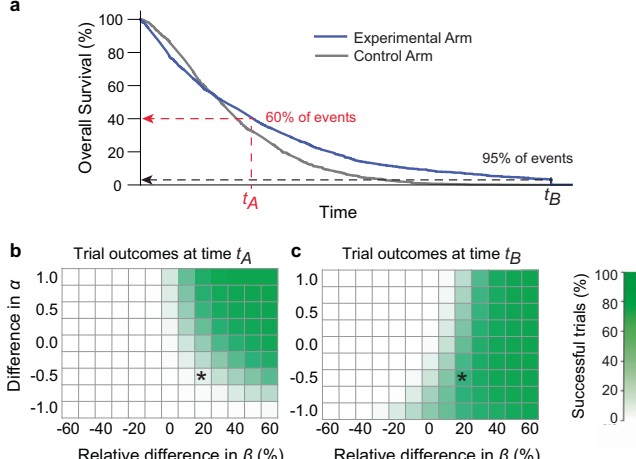

**Fig. 7 Effect of trial duration on success when proportional hazards are violated. a** One of many simulated trials having a range of Weibull $\alpha$ and $\beta$ parameters similar to those observed in actual trials reporting OS data; in this trial, $\Delta\alpha = -0.5$ and the ratio of $\beta$ for experimental and control arms was 1.2. The labels $t_A$ and $t_B$ denote times corresponding to approximately 60% or 95% of all trial events (for real OS trial arms in this article, in metastatic cancers, these event rates correspond to median times of $t_A = 22$ months and $t_B = 50$ months). **b** Percent of simulated successful trials at time $t_A$ or **c** $t_B$. The simulated trial depicted in panel **a** is denoted by an asterisk. "Success" was scored as a hazard ratio less than one at a 95% confidence level using the Cox proportional hazards regression; this metric was used despite the violation of proportional hazards because it is the accepted approach for assessing efficacy in pivotal trials (see text and "Methods" for details).

this assumption is frequently violated[13,15,47]. From the perspective of Weibull distributions, proportionality means that the two arms have the same shape parameter (i.e., $\Delta\alpha = 0$ where $\Delta\alpha$ is the difference in $\alpha$ values). A trial with $\Delta\alpha = 0$ is successful if the experimental arm has significantly larger $\beta$ value than the control arm. However, across 121 comparisons of experimental and control arms from 116 OS trial figures, we found that $\Delta\alpha$ values actually varied from +0.65 to −0.80 (median absolute value $|\Delta\alpha| = 0.11$; Fig. 6a). For event-free survival data, 155 comparisons of experimental and control arms from 146 trial figures revealed a range of $\Delta\alpha$ values from +0.55 to −0.85 (median $|\Delta\alpha| = 0.08$) (Supplementary Fig. 9). Using a traditional Grambsch–Therneau test[15,48], a proportional hazards violation was found in 18/108 OS and 47/135 event-free survival trials, as well as 8/10 ICI trial comparisons (3/5 for OS and 5/5 for PFS)[15]; a corresponding 90% confidence level yields a $|\Delta\alpha| = 0.30$ threshold for significant violations of proportional hazards.

To explore the origins and consequences of non-proportionality of hazards in survival data we used best-fit Weibull shape and scale parameters to calculate the ratio of cumulative hazards at time $t$ ($HR_c(t)$), an approach that makes no assumptions about proportionality, and compared $HR_c(t)$ to the hazard ratio calculated using Cox regression ($HR_{SP}$; which is semiparametric and time invariant). In both successful and unsuccessful trials for which $|\Delta\alpha|$ was small, $HR_c(t)$ (blue lines in Fig. 6b, c) closely approximated $HR_{SP}$ (red lines). In contrast, when $\Delta\alpha = -0.26$ (for the successful trial CheckMate 057; NCT01673867[49]) $HR_c(t)$ and $HR_{SP}$ often differed (Fig. 6d). This was also true of CA184-043 (NCT00861614; $\Delta\alpha = -0.30$), which was judged to have failed based on $HR_{SP}$[50](Fig. 6e). However, in this trial, $HR_c(t)$ fell steadily over time and had reached a value below one by the end. Unless the shape of the hazard function

were to change substantially after month 25, it seems probable that CA184-043 would have been judged a success had it continued for only a few months longer.

To more fully explore the dependence of trial duration on outcome, we simulated a series of two-arm trials having a range of differences in $\Delta\alpha$ and $\Delta\beta$ values. We then plotted the fraction of trials that were successful, as judged by Cox regression. Success was evaluated both at an early stopping point, when ~60% of events had been recorded ($t_A$) or a late stopping point when ~95% of events had been recorded ($t_B$; Fig. 7; "Methods"). For simulated trials in which $\alpha$ was smaller for the experimental than the control arm, the experimental arm exhibited lower survival at early times and then crossed over the control arm at later times (as shown in Fig. 7a). In these cases, a later time point was associated with a greater likelihood of success than an earlier time point (Fig. 7b, c). The greater the value of $|\Delta\alpha|$, the greater the impact of curve crossing and duration of follow-up on outcome. Moreover, OS results from all ICI trials in our data set fell into this category (e.g., Fig. 6a). The reasons for time-dependent therapeutic effects are unknown, but in ICI trials it has been suggested that they arise from treatment-related toxicity at early times or delayed treatment effects[13,51,52].

The importance of trial duration on success is demonstrated by the MK-3475-022/KEYNOTE-022 trial of pembrolizumab with dabrafenib and trametinib for *BRAF*-mutant melanoma. The pre-planned analysis at 24 months did not identify a statistically significant benefit (PFS hazard ratio of 0.66, 95% CI: 0.40–1.07) but a subsequent analysis at a median 36.6 months did (PFS hazard ratio of 0.53, 95% CI: 0.34–0.83)[41,53]. Parametric fitting of the original Phase 2 data at 24 months found a difference in the survival curve $\alpha$ values ($\Delta\alpha = -0.21$). This is the scenario in which a statistically significant benefit from therapy is more likely to be identified at longer follow-up, as was confirmed by data at 36.6 months.

From these data, we conclude that oncology trials exhibit continuous deviations from the assumptions of the proportional hazards model. The underlying variation in treatment effect over time can be identified by Weibull fitting as situations in which $|\Delta\alpha| \gg 0$. In these cases, the duration of the trial can have an effect on the likelihood of success in a manner that is not accounted for by Cox regression. We suggest that future trials, particularly of ICIs, evaluate $\Delta\alpha$ and model the possible impact of trial duration on the likelihood of success.

## Discussion

Using a set of ~220,000 imputed participant survival events from published oncology trials we find that survival functions for solid tumors, including those from trials that report OS or event-free survival data (e.g.: PFS) or are biomarker-stratified, are well fit by two-parameter Weibull distributions. The poorest fits are often explainable by pre-planned changes in treatment and by the confounding effects of radiological scan times on evaluation of PFS. The Weibull $\alpha$ (or shape) parameter defines increasing or decreasing hazard over time and the $\beta$ parameter is proportional to the median survival time, making fitted parameter values readily interpretable. Both $\alpha$ and $\beta$ differ between treatment and control arms; $\Delta\alpha$ quantifies violations in the assumption of proportional hazards that is used in Cox regression and $\Delta\beta$ measures the magnitude of the therapeutic effect. The excellent fit of survival data to a single parametric function for many types and stages of cancer and across drug classes demonstrates that therapeutic benefit can be well-described by a simple function in which responses vary continuously across a population. In the trials studied here, the likely presence of prognostic factors, or responder and non-responder populations, did not sufficiently

separate survival functions to produce bimodal distributions that would have necessitated the systematic use of mixture models or cure-rate parameters.

Our findings support modeling survival in early stage oncology clinical trials by using parametric statistics. Parametric statistics are already used in cost-effectiveness analysis[21] and other simulation studies, although it has not been established which parametric forms (Weibull, Log-Normal, Gompertz–Makeham, etc.) accurately fit empirical survival data. We now establish that Weibull distributions involving clinically interpretable parameter values have sufficient accuracy to be the preferable parametric form for describing survival in trials of solid tumors. By simulating trials of different sizes, we find that modeling with parametric methods substantially improves precision with equivalent accuracy: point estimates of survival (e.g., at 12-months) using a Weibull-based parametric approach approximately doubles the number of trials in which informative confidence intervals can be obtained. In cases in which parametric and nonparametric approaches can be compared directly, we find that a 50-patient trial arm reporting OS data is as precise as a 90-patient trial arm evaluated using nonparametric methods. This advantage pertains primarily to trials with small numbers of participants (20–100 per arm); when arms are larger, conventional Cox regression is the preferred method. Thus, the use of parametric methods based on Weibull distributions should be strongly considered in early phase signal-seeking studies with the goal of rapidly and economically identifying the optimal setting in which to perform Phase 3 trials.

Weibull distributions are also appropriate for cost-effectiveness research for oncology drugs, an increasingly important topic for drug approval in many countries. In this context, it is important to note that Weibull and Log-Normal distributions provide equivalently good fits to IPD and the Weibull form was chosen in the current study because of its interpretability in terms of hazard rates and median survival. Log-Normal distributions may have corresponding advantages in pharmaco-economic analysis[54]. Moreover, insofar as there exist multiple ways to implement parametric statistics, we note that our results pertain specifically to approaches detailed in the "Methods", which are conventional. Alternative approaches to increasing trial precision, for example by changing significance levels to create narrower nonparametric confidence intervals while still maintaining an acceptable level of Type I error, have not yet been empirically explored in detail but can be pursued using the imputed IPD provided with this manuscript.

In current practice, Cox regression is used to compare survival functions based on the proportional hazards assumption, which states that the ratio of control and experimental hazard rates is constant over time. Success usually corresponds to a hazard ratio less than one at 95% confidence. With respect to Weibull distributions, the assumption of proportional hazards corresponds to no difference in shape, i.e., $\Delta\alpha = 0$. It is well established that a subset of trials deviate from the proportional hazards assumption[13,15]. However, we find that $\Delta\alpha$ varies over a wide range, from ~+0.7 to −0.8, and that the majority of trials analyzed deviate from the proportional hazards assumption to some degree. If we apply previously described criteria (the Grambsch–Therneau test at a 10% threshold)[48] to identify significant deviations from proportional hazards[13,15] we find violations in ~17% of trials reporting OS and ~35% of trials reporting PFS data, and that this significance level corresponds to $|\Delta\alpha| > 0.3$.

Analysis of imputed data from published trials and simulations using empirical survival functions shows that violations of proportional hazards arise from time-varying treatment effects. In the data analyzed here, this was most evident in ICI trials, but was also seen in trials of the BCL-2 inhibitor venetoclax, in which experimental and control arm PFS curves cross each other after

trial initiation[55]. The biological basis of time-varying treatment effects (and curve crossing) are not known but could arise from high toxicity in a subset of patients early in treatment, delayed onset of treatment effects, or exceptionally durable responses in some patients (indicating the presence of prognostic factors)[13,15,51,52]. Regardless, the practical consequence of these effects is that the duration of a trial has a direct impact on outcome, independent of statistical considerations such as increasing confidence in hazard ratio values as trial events accrue.

We found that it was possible to use Weibull fitting to identify trials judged as failures by Cox regression in which the ratio of cumulative hazards was trending steadily below one at the end of a trial, and an extension of only a few months was predicted to result in success. Future work could explore the use of Weibull fitting in trial interim analysis, particularly in trials where $|\Delta\alpha| \gg 0$ such as for ICIs, to help determine when to terminate trials for treatment futility. Additional changes that could be implemented in future trials include improving how sample size and power are estimated. Such calculations are most commonly performed under an assumption of an exponential fit to survival data. Alternative distributions have been proposed for such analyses[56–59] but without any means for selecting optimal parameter values for simulation. Using the Weibull fitting described here, empirically-derived parameter values can be drawn directly from past trial data. A final set of applications involves the use of parametric forms for subgroup analysis in Phase 3 and basket trials. Since studies of this type are intended to test therapeutic hypotheses rather than lead to drug registration, the regulatory barriers to using parametric methods are limited. Parameterized Bayesian trial designs, such as the continual reassessment method (CRM) or escalation with overdose control (EWOC), are other model-based methods already in use to define specific parameter values and improve the efficiency of Phase 1 studies[60].

Even modest improvements in the design and interpretation of oncology clinical trials are likely to have a substantial payoff. The overall approval rate for new oncology drugs remains low: only 3% of drugs tested in a Phase 1 clinical trial and 7% of drugs tested in a Phase 2 study are ultimately found to be superior to standard of care comparators in pivotal Phase 3 studies and approved[61]. Methods to more accurately understand drug activity in small patient populations are included in the National Cancer Institute's (NCI) 2020 "provocative questions" and could lead to a wider use of master protocol trials. Improving trial efficiency and predictability will become increasingly critical as the number of new monotherapies and combinations continues to rise, patient populations become more subdivided based on the molecular characteristics of their tumors, and it becomes impractical to enroll enough patients to test all promising drug treatments[62].

The use of nonparametric statistics was historically appropriate because treatment effects could be calculated precisely without the need for extensive computation[46], which was largely infeasible prior to the widespread availability of personal computers. Moreover, the proportional hazards assumption appears to be largely valid when assessing OS in the context of cytotoxic chemotherapies (OS trials including chemotherapies in our data set had a median $|\Delta\alpha| = 0.10$, well below the $|\Delta\alpha| = 0.30$ threshold for significant violation). However, the deviation from proportional hazards reported in this and previous studies, and its likely origins in the biology of new and more diverse forms of cancer therapy, call for a reconsideration of Cox regression. Several approaches for comparing treatment effects have been proposed including weighted[63] or adaptive log-rank tests[64], restricted mean survival times[65,66], and permutation-based approaches[67]. Trialists, sponsors, and regulatory agencies may want to examine the use of such methods in the setting of the ICI trials that are the focus of so much current research.

**Limitations of this study**. This work is not a formal meta-analysis or systematic review of a specific treatment regimen or disease, but instead a broadly conceived research study; no treatment decisions should be made based on our findings. A specific limitation is that we only analyze four tumor types (breast, colorectal, lung, and prostate); extending the analysis to other cancer types will require imputing IPD from additional trials. Additionally, we use Cox regression to determine whether real or simulated trials are "successful" (e.g., Figs. 6 and 7) even when their survival distributions clearly violate proportional hazards. We do this because a hazard ratio less than one at a pre-specified level of confidence is the only widely accepted method for evaluating trial outcomes. Finally, we used subsampled patient events from completed, Phase 3 trials to infer the properties of small patient populations commonly used in Phase 1 and 2 trials on the assumption that underlying survival functions can be described by similar parametric forms in early and late stage trials. We are unable to rigorously assess this assumption but we find that simulated trials comprising four different cancer types are also well fit by two-parameter Weibull forms, suggesting that having a heterogenous patient population (as encountered in many early phase trials) does not reduce the accuracy of para-metric analysis. The parameters of best-fit Weibull distributions are very likely to differ between Phase 1 or 2 and Phase 3 studies whenever there are differences in inclusion criteria, such as prior therapy, performance status, tumor stage, and histology. Notably, a similar limitation can be expected from other methods (e.g., traditional nonparametric approaches) that use early-phase efficacy data to design pivotal studies.

A final limitation of this study is that it uses imputed IPD rather than original data. We are forced to do this because original results are simply not released, and most published oncology trial reports do not provide the numerical values used to plot the Kaplan–Meier estimators. The release of numerical data underlying graphical representations has become the norm in pre-clinical research and is at the heart of efforts by funding agencies to make data FAIR (findable, accessible, interoperable, and reusable). Multiple calls have been made to make IPD from research clinical trials publicly accessible to ensure the reproducibility of study results and facilitate meta-analyses, but compliance remains low[12]. Outside of oncology, calls for reuse of both contemporary and historical control arms have arisen in repurposing trials for COVID-19, particularly when the same set of institutions is conducting many parallel trials outside of a master protocol framework. Recent work has attempted to identify factors contributing to this non-compliance, which include the time involved in data annotation, an absence of standardized provisions for sharing IPD, and concerns about patient privacy[68]. We have made the data described in this paper available via an interactive website (https://cancertrials.io/) that we will continue to expand with new imputed data and analysis. To reduce barriers to sharing IPD, trialists are invited to post their primary data on cancertrials.io (contact information on the website).

Ongoing data collection efforts relevant to clinical trials include Project Data Sphere[69], the NCI's National Clinical Trials Network (NCTN) and Community Oncology Research Program (NCORP) Data Archive, and The Yale University Open Data Access (YODA) Project[70]. Unfortunately, these projects have substantial limitations with respect to the type of analysis presented here: (i) most IPD are greater than six years old and do not cover many of the drugs of greatest current interest, including ICIs; (ii) most public data derives from control, not experimental treatment arms; (iii) much of the data involves summary statistics, not IPD, and requests for underlying data can be strictly limited; (iv) if access to IPD is granted, they are often available online for inspection but are not downloadable for additional computational analysis. A substantial unmet need, therefore, exists for primary data from clinical trials to be made available for reuse. One approach is to amend the requirements for data deposition on ClinicalTrials.Gov (per U.S. Public Law 110-85) to include IPD.

## Methods

**Individual participant data imputation and curation**. The original data set consisted of 153 unique trials in breast, colorectal, lung, and prostate cancer in the metastatic and non-metastatic settings from 2014 to 2016[22]. Trials were removed from the original data set if there were any inconsistencies in the imputed patient data as compared to its associated clinical trial (e.g.: differing numbers of patients from the publication at-risk table and imputed data). The quality of the data imputation was confirmed quantitatively, by calculating the hazard ratio for imputed data and comparing it to the corresponding trial's reported hazard ratio, and qualitatively, by overlaying the Kaplan–Meier curve generated from the imputed data on top of the published curve. Trials with a hazard ratio difference greater than 0.1, or with perceptible visual differences, were removed from the final data set and not analyzed further (Supplementary Data 1).

**Parametric fitting of patient survival data**. The imputed event times, either death for overall survival distributions or surrogate events in the case of event-free survival distributions, were compared to the event times simulated under each parametric distribution. The likelihood of a specific parametric form to fit patient data was computed by maximum likelihood estimation. Specifically, the relative likelihood of a patient event taking place at a particular point in time was calculated under that parametric distribution's probability density function. The likelihood of a censoring event taking place was calculated by integrating the probability density function (the CDF), and computing the likelihood of a patient event taking place in the trial after the censoring time (1-the cumulative probability up to that time). This procedure was repeated for all patient events in an arm of a clinical trial, and the overall likelihood of a fit was calculated by multiplying all relative likelihoods.

**Computing $R^2$ explained by the Weibull fit**. For imputed patient events in a clinical trial arm, the event times (deaths or surrogate events) and corresponding percent survival (OS or event-free survival) were computed. Weibull parametric fitting was used to obtain the best-fit $\alpha$ and $\beta$ values corresponding to the imputed patient data. The differences between the survival distribution under a best-fit Weibull model and the imputed data were analyzed through a Weibull plot[29]. In this approach, the event times and corresponding survival are normalized such that if the data follow a Weibull distribution, the points will be linear. The event times were normalized through the transformation: $\ln t/\beta$, while survival was normalized by: $\ln (-\ln S(t))/\alpha$. Coefficient of determination ($R^2$) values were calculated to assess the goodness of Weibull fitting for all trial arms in the data set.

**Computing Weibull fits to trials of immune checkpoint inhibitors**. Trials of ICIs were selected from the data set (five in total). Each trial's OS and PFS IPD were fit to a single Weibull distribution and a mixture distribution made of two Weibull distributions. An additional set of simulations was performed to account for the periodicity of radiological scans in detecting progression events. The quality of fit for these simulations is not readily interpretable through use of a Weibull plot, and was instead quantified by the coefficient of determination ($R^2$) between observed and fitted PFS.

**Assessing the relationship between trial scan times and PFS drops**. Trials of ICIs in oncology were obtained through a PubMed search of the terms "neoplasms" or "cancer" and "Clinical Trial, Phase III" along with therapies of interest ("ipili-mumab" or "pembrolizumab" or "nivolumab"). The search was filtered to yield 25 trials with PFS data and a reported scan time, in addition to the five trials in the original data set, for a total of 30 trials for subsequent analysis. PFS curves were extracted from each of the trials and images were analyzed using DigitizeIt software (version 2.5.3; Braunschweig, Germany) to estimate the timing of the PFS drop in each survival curve. The trial scan time interval was obtained from each publication's methods, blinded to the image associated with each trial. All extracted values can be found in Supplementary Data 3.

**Simulating differences in trial success based on $\alpha$ and $\beta$ values**. Control and experimental arms of clinical trials were simulated 1000 times by drawing 100 patient events from Weibull distributions with differing $\alpha$ and $\beta$ values. $\alpha$ values in the experimental arm ranged from 0.5 to 4.5, $\beta$ values from 0.4 to 1.6, and control arm parameter values were kept constant (1.5 and 1 respectively). Figure 7 shows results in the region of interest, from experimental arm $\alpha = 0.5$ to 2.5. Events were censored at either early time points (corresponding to a ~60% event rate, a time equal to the control arm $\beta$ value) or later time points (corresponding to a ~95% event rate, a time equal to four times the control arm $\beta$ value). Trial success was calculated for each simulation by using a Cox regression at a 95% confidence level. Significance was calculated using a Wald test.

**Calculating the precision and accuracy of survival estimates**. IPD for each trial arm in the data set was extracted. 213 OS and 273 event-free survival trial arms were used for further analysis; these trial arms had at least 100 patients (the maximum number of patient subsampling events used in this experiment) and at least one event (i.e.: death, progression) taking place before 12 months. For each trial arm, 20–100 patient events (with a step size of 10 events) were subsampled from the imputed IPD. At least three non-censoring events were selected during each sampling simulation. This procedure was repeated ten times per sample size and trial. Parametric and nonparametric 95% confidence intervals for 12-month survival were computed for every sampling simulation.

Accuracy and precision plots were constructed for the subset of simulated trial arms returning numerical nonparametric confidence intervals (125 OS trial arms and 99 event-free survival trial arms). Note that nonparametric estimates did not return a numerical confidence interval for 41% of OS trial arms and 64% of event-free survival trial arms, while Weibull fitting made it possible to calculate 12-month confidence intervals for every trial arm in every simulation.

**Quantification and statistical analysis tools**. Analysis was performed using Wolfram Mathematica Version 12.1. Details of the statistical analysis performed, exact values of $n$ and what they represent, definitions of the summary statistics used, definitions of significance, and trial inclusion and exclusion criteria can be found in the "Method" details, Figure captions, and "Results" sections of the manuscript. Compute-intensive analyses (e.g., sample size simulations) were conducted on the O2 High Performance Compute Cluster, supported by the Research Computing Group, at Harvard Medical School.

**Reporting summary**. Further information on research design is available in the Nature Research Reporting Summary linked to this article.

## Data availability

All data generated or analyzed during this study are included in this published article (and its supplementary information files). Data are also available through the website https://cancertrials.io/ and Synapse (ID: syn25813713).

## Code availability

All code used in this study is included in Supplementary Data 2. Code is also available through Synapse (ID: syn25813713). Each piece of code is provided in a folder containing a Mathematica Notebook (.nb), all data required by the code, and the corresponding code output. With source data kept within the same folder as the code, the Mathematica Notebook can be executed in Wolfram Mathematica by selecting "Evaluate Notebook" from the "Evaluation" menu. Sample R code (R version 4.0.3) illustrates the parametric fitting and confidence interval construction procedures. Pseudocode files summarize the algorithms used to execute analysis corresponding to each result.

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

## Acknowledgements

We thank Lorenzo Trippa, Andrea Arfè, Giovanni Parmigiani, John Higgins, Charles Perou, Michael Kosorok, Haeun Hwangbo, Nicholas Clark, Clemens Hug, and Claire Lazar Reich for their helpful comments on this project. We thank Jeremy Muhlich and Zev Ross for their assistance in implementing the https://cancertrials.io/ site. We thank William Sorger for his assistance in recording the trial metadata. We are grateful to all of the patients and investigators who participated in the clinical trials analyzed in this work. This project was supported by NIH grants P50-GM107618 and U54-CA225088 (to P.K.S.). D.P. is supported by NIGMS grant T32-GM007753 and F30-CA260780.

## Author contributions

Generating individual participant data set: G.F. and B.M.A. Metadata curation; data analysis: D.P. and A.C.P. Writing: D.P., A.C.P., and P.K.S. Manuscript review and editing: D.P., G.F., B.M.A., A.C.P., and P.K.S. Supervision: B.M.A., A.C.P., and P.K.S. Funding: B.M.A. and P.K.S.

## Competing interests

P.K. Sorger is a member of the SAB or Board of Directors of Applied Biomath, Glencoe Software, RareCyte Inc and has equity in these companies; he is a member of the SAB of NanoString Inc and a consultant for Merck and Montai Health. In the last five years the Sorger lab has received research funding from Novartis and Merck. Sorger declares that none of these relationships are directly or indirectly related to the content of this manuscript. B.M. Alexander is an employee of Foundation Medicine. No potential competing interests were disclosed by the other authors.

## Ethics approval

All trials analyzed in this manuscript have been previously published (list found in Supplementary Data 1). Any information that could lead to the identification of an individual patient was not accessed, and no concerning ethical issue was raised in this research that would necessitate ethical approval or participant consent.

## Additional information

**Peer review information** *Nature Communications* thanks Ying Yuan and the other anonymous reviewer(s) for their contribution to the peer review this work. Peer reviewer reports are available.

