## [Peer Review File · Nature Communications]

Reviewers' Comments:

Reviewer #1:

Remarks to the Author:

This paper imputed survival IPD from 500 phase III oncology trials including ~220K events and claimed Weibull distribution fits the survival curves well. Thus, the authors recommended to use parametric modelling based on Weibull distribution, which will increase the power. They also pointed the immunotherapy studies needs to use cure rate model which violate the proportional hazard assumption; thus the traditional Cox proportional hazard modeling is not the best approach, and a three-parameter Weibull distribution fits better. They use image processing to impute IPD from published KM and also generated a website cancertrials.io to illustrate the survival curves and Weibull distributed curves. The overall presentation is clear. However, several improvements can be made:

1. Weibull distribution fits oncology trial curves well is not novel. It might be good to discuss why parametric modeling is less popular than either semi-parametric modeling Cox or nonparametric KM. Also, it's not clear whether parametric modeling based on Weibull have appropriate statistical properties to do regression with one or more covariates, similar as Cox. Btw, to help readers to use parametric modeling, I would recommend to add a simple example code in R (Mathematica is not free and not open source, right?).
2. Some statement needs more discussion. For example, "nonparametric estimates did not return a numerical confidence interval for XX, while Weibull fitting made it possible to calculate XX". KM, Cox, and Weibull can all generate appropriate confidence interval, and I assume the points they would like to claim is "Not all published trial reported confidence interval by nonparametric method" and should discuss why many studies can but didn't provide this important information.
3. Most of the trials data used here are phase III oncology with larger sample size, but the recommendation is to use parametric method on phase I/II studies with small sample size because it will improve the power. Supp Figure 5 is a good example for small sample size. Btw, Figure S5 A showed Nonparametric method has better accuracy than Parametric method when n increases, right? Recommend to add some comments on why.

Reviewer #2:

Remarks to the Author:

This paper imputed survival data from 500 arms of phase III oncology trials, and found that the data can be well fitted by Weibull distribution. This is an interesting and potential useful work.

1. One main point of the paper is that the result suggests that better inference (e.g., narrower CI) can be made for small phase I and II trials using Weibull distribution. This, however, depends on the assumption that the distributions of PFS and OS in phase I and II are the same as those in phase III, which may not be true. Phase I and II populations are often much more heterogenous than phase III. This extrapolation may be problematic.
2. A key metric the paper used to report and support the result is R2. For survival data, R2 is not widely used because it is not as interpretable as in linear regression. The accurate definition of R2 should be provided, as well as its interpretation for survival data.
3. P6, the second paragraph, given the observed R2 is 0.981 and the theoretical maximum R2 is 0.995, the interpretation that "Thus, the error between observation and the Weibull model is ~1%" is incorrect. R2 does not have such interpretation.
4. P6, it is useful to show the boxplot of R2 for 237 trial arms, including median, quantiles, in addition to the pooled mean.
5. The paper focuses on the marginal distribution of OS and PFS. It should be noted that the conditional distribution may not be Weibull. In practice, the main motivation of using nonparametric and semiparametric methods is to minimize assumptions and product robust results.
6. The paper spent 3 pages to discuss the violation of proportional hazards. This fact is well known, with extensive publications, especially for ICI. I do not see the need of extensive discussion of this under Weibull model, and is remotely related to the main result (PFS and OS can be well fitted by Weibull distribution). This section should be substantially shortened (<1 page).

DETAILED RESPONSE TO REVIEW

MS# NCOMMS-21-22949-T

Updated title “Cancer patient survival can be accurately parameterized, improving trial precision and revealing time-dependent therapeutic effects.”

We thank the reviewers for a helpful set of suggestions. In response, we have made changes throughout the manuscript, generated code for survival data fitting in R, and added new analysis and new supplementary figures. We have also modified existing figures and supplementary materials to meet the journal guidelines, and we are submitting completed reporting, software, and editorial policy checklists.

We believe these changes address all of the reviewers concerns and have resulted in a clearer and more comprehensive manuscript. Specific changes can be found as tracked changes in the text.

Reviewer #1 (Remarks to the Author):

This paper imputed survival IPD from 500 phase III oncology trials including ~220K events and claimed Weibull distribution fits the survival curves well. Thus, the authors recommended to use parametric modelling based on Weibull distribution, which will increase the power. They also pointed the immunotherapy studies needs to use cure rate model which violate the proportional hazard assumption; thus the traditional Cox proportional hazard modeling is not the best approach, and a three-parameter Weibull distribution fits better. They use image processing to impute IPD from published KM and also generated a website cancertrials.io to illustrate the survival curves and Weibull distributed curves. The overall presentation is clear. However, several improvements can be made:

Before addressing the reviewer’s important conceptual and technical points (see below) we do want to point out that we do not propose that pivotal Phase 3 trials use a three parameter cure-rate model as part of the primary analysis and subsequent regulatory decisions. Instead, we propose that parametric fitting based on Weibull forms be used for early phase trials having fewer than ~100 patients. These trial results are widely used in prioritizing and designing Phase 3 studies. For example, in the case of ICI trials this includes defining what duration of follow-up is most likely to reveal a significant survival benefit (this is also a setting in which non-proportional hazards can complicate the discovery of benefit at early time points). Moreover, we wish to emphasize that we strongly favor the use of two-parameter Weibull forms (not three-parameter forms) that only have shape and scale parameters. This is a more parsimonious way of describing large amounts of data and we see no need, from the perspective of goodness of fit, for introduction of a third “cure rate” parameter. We have modified the text in multiple places to make these points clear.

We speculate that the success of pivotal trials will be improved by better understanding early phase data, but this does not require or benefit from the use of parametric statistics with large patient populations in pivotal trials (a change that is not supported by our data and likely to be far harder to implement).

1. Weibull distribution fits oncology trial curves well is not novel. It might be good to discussion why parametric modeling is less popular than either semi-parametric modeling Cox or nonparametric KM. Also, it’s not clear whether parametric modeling based on Weibull have appropriate statistical properties to do regression with one or more covariates, similar as Cox. Btw, to help readers to use parametric modeling, I would recommend to add a simple example code in R (Mathematica is not free and not open source, right ?).

We thank the reviewer for the suggestion that we provide present context for our work; we have therefore revised the Introduction and Results (Page 3, paragraph 3; Page 5, paragraph 2). Specifically, the revised Introduction now emphasizes the reviewer's point that parametric fitting of survival data in medicine and engineering is far from novel, and is in fact commonly performed when quantifying the expected benefit of an intervention in econometric analysis. A wide variety of parametric forms have been proposed for survival data including Log Normal, Log Logistic, Gamma, Weibull, Gompertz–Makeham, and Exponential distributions. The accuracy of these fitting procedures has until now been a matter of conjecture: in the absence of large-scale digital survival data it has not been possible to evaluate goodness of fit. Thus, most theoretical and comparative effectiveness research is limited by the absence of information on which parametric form represents the best match to reality. The primary innovation in our study is to evaluate goodness of fit of a Weibull distribution to a large number of trials and to investigate the poorest fits in detail. We find that poor fits are frequently associated with pre-planned changes in protocol and - in the case of PFS – with confounding by scan time. Overall, our analysis demonstrates that two-parameter Weibull forms are an informative and accurate parameterization for survival data in solid tumors.

In the absence of an evaluation of fitting accuracy, it is entirely appropriate that parametric methods have been less popular than semiparametric or nonparametric methods. The reviewer will appreciate that the hard part has been getting the data into a usable form – not performing the fitting per se. Our digitization makes it feasible to use a data-driven approach to quantify the relative fit of different survival distributions to a large, heterogeneous, set of cancer patient data. We also provide a description of settings in which use of a Weibull distribution is appropriate and why it is advantageous in these settings (e.g. increased precision for small sample sizes). We also discuss settings in which use of current methods remains appropriate (large trials).

Estimating covariate effects is not an intended application of the analyses of this study. As a practical matter, imputed patient data lack covariate annotations, and thus, testing the validity of parametric methods for such an analysis is not possible using currently available data. This is unfortunate but if our work encourages the authors of trial reports to make covariate annotations available, then it will be possible to delve into a host of interesting questions related to covariates.

Covariate effects are typically estimated using large, well-powered, Phase 3 trial data, a setting that – based on current understanding – is unlikely to benefit from use of a parametric form for estimating efficacy. Phase 3 trials would benefit, however, from alternative to the proportional hazards model used in Cox regression, and more careful consideration of the impact of trial duration on an assessment of benefit. It is possible that if covariates sufficiently subdivide a Phase 3 dataset (into small patient groups) then parametric statistics will be useful; however, we are not yet in a position to evaluate this possibility.

We hope that research such as ours demonstrating the value of secondary analysis may encourage sharing of primary data in the future, and we have revised our Discussion to call attention to this important limitation in current data-sharing practices. As mentioned above, were the necessary primary data available, including demographic information, future work could use parametric fitting to quantify covariate effects (as frequently done by semiparametric approaches in Phase 3 trials) and also assess whether the shape (α) of survival distributions is affected by or the assumption of proportional hazards is violated by covariate analysis in these settings (e.g.: breast cancer tumor grade; see Bellera et al., BMC Medical Research Methodology, 2010).

We agree with the reviewer that non-proprietary software is important and we have therefore included R code, with a sample instantiation of the procedure, as part of Supplementary Material 2. We appreciate this thoughtful suggestion. We are also continuing to improve our web site (which is based on R-Shiny) so it is easier to find and download subsets of the data.

2. Some statement needs more discussion. For example, “nonparametric estimates did not return a numerical confidence interval for XX, while Weibull fitting made it possible to calculate XX”. KM, Cox, and Weibull can all generate appropriate confidence interval, and I assume the points they would like to claim is “Not all published trial reported confidence interval by nonparametric method” and should discuss why many studies can but didn’t provide this important information.

We believe that our wording on this point was confusing and apologize. Nonparametric methods are unable to report a numeric confidence intervals when too few events occur before or after the time point at which the analysis is being conducted; this occurs infrequently in large, well-powered Phase 3 trials, but commonly in trials with small samples. This is illustrated in a figure below, which is also included as a new supplementary figure (Supplementary Fig. 5). “Non-informative” estimates correspond to subsampled survival curves with 0% (left panel) or 100% survival (right panel) at 12 months. We have made extensive revisions to this part of the text to address this concern,

3. Most of the trials data used here are phase III oncology with larger sample size, but the recommendation is to use parametric method on phase I/II studies with small sample size because it will improve the power. Supp Figure 5 is a good example for small sample size. Btw, Figure S5 A showed Nonparametric method has better accuracy than Parametric method when n increases, right ? Recommend to add some comments on why.

The reviewer is correct: almost all analysis presented in the manuscript was performed on Phase 3 clinical trial data. We did this because well-powered Phase 3 trials provide a “ground truth” against which to assess the accuracy of parametric fitting: with 200 to 1000 patients per arm, their survival distributions are empirically well-determined. Phase 1 or 2 trials entail smaller cohorts and much larger margins of uncertainty and as practical matter, relatively few report survival distributions. We have

nonetheless examined Phase 2 distributions that are available (e.g. KEYNOTE-022) and do not observe any systematic deviation from a Weibull form.

We describe below (point 1 in response to reviewer 2) a new simulation we performed to determine if Weibull fitting is appropriate for analysis of the heterogenous patient populations sometime encountered in Phase 1 and 2 trials.

*As the reviewer correctly states, as sample size increases, the accuracy of nonparametric methods appears to surpass that of parametric methods. This is due to an unavoidable limitation of our approach: ground truth in our simulations is **defined** as the nonparametric survival for the full Phase 3 trial. Therefore, by definition, as sample size increases, the error of the nonparametric approach reaches zero. Unfortunately, in the absence of any other established means of determining ground truth for a full Phase 3 trial it is not possible to fairly compare parametric or non-parametric methods. We thank the reviewer for noticing this important point. We have revised the Results section to include the explanation presented above (page 11, paragraph 2). Of note, as discussed above, our findings do not support the use of parametric methods to estimate drug efficacy from Phase 3 trial data.*

Reviewer #2 (Remarks to the Author):

This paper imputed survival data from 500 arms of phase III oncology trials, and found that the data can be well fitted by Weibull distribution. This is an interesting and potential useful work.

1. One main point of the paper is that the result suggests that better inference (e.g., narrower CI) can be made for small phase I and II trials using Weibull distribution. This, however, depends on the assumption that the distributions of PFS and OS in phase I and II are the same as those in phase III, which may not be true. Phase I and II populations are often much more heterogenous than phase III. This extrapolation may be problematic.

We agree with the reviewer's point and have extensively revised the Discussion to emphasize that the current analysis supports use of Weibull parameterization when a small patient cohort resembles a subset of a large patient cohort. This is of course true of both parametric and non-parametric methods. Early-phase and late-phase trials often have differences in patient characteristics that affect survival with and without therapy. We have revised the discussion to emphasize that it is important not to overlook differences in patient characteristics in early-late comparison and that we have shown our approach to be valid only with Phase 2 trials that are not substantially more diverse than is typical at Phase 3 with respect to tumor histology, stage, and prior therapy.

The more specific concern is whether early phase trials can reasonably be parameterized using two-parameter Weibull forms. One argument in favor of this is that the Phase 3 trials we have analyzed involve a variety of tumor types, stages (local or advanced), and prior treatment (naive or pre-treated); in all cases Weibull forms exhibited excellent fits.

The question nonetheless arises whether Phase 2 cohorts in general have a mixture of these characteristics (e.g. in multi-histology 'basket' trials) that require different parameterization. A practical challenge in testing this (also explained in response 1.3. above) is that Phase 3 trials are the only valid way to assess the accuracy of a parametric fitting method because they provide a well-powered "ground truth." We have found relatively few examples of matched Phase 2 and 3 survival curves in the literature, although we presume that trial sponsors have this information.

As an alternative approach we simulated trials having very heterogeneous patient populations comprising a random mix of four different tumor types (breast, colorectal, lung, and prostate) by randomly sampling events from Phase 3 overall survival data. We then tested goodness of fit to Weibull parametric forms. When we looked at metastatic disease alone (172 OS curves; right panel below) we found that two-parameter Weibull forms accurately described the simulated data ($R^2=0.98$)* and addition of a third “cure rate” parameter barely improved this (to $R^2=0.99$; red line). When metastatic and non-metastatic disease were combined using data from 237 OS curves (left panel below) however, a two-parameter Weibull form exhibited $R^2=0.95$ and addition of a third “cure rate” parameter improved the fit to $R^2=0.98$ (red line).

These simulations support the use of two-parameter Weibull forms with small patient populations having heterogenous types of cancer at a similar stage. We have included this point in the text and in a new Supplementary Figure 7. However, we still favor two parameter forms because we think it unlikely that metastatic and non-metastatic tumors would unknowingly be mixed in a clinical trial.

*See point 2.2. for explanation on the definition of R^2

2. A key metric the paper used to report and support the result is R^2 . For survival data, R^2 is not widely used because it is not as interpretable as in linear regression. The accurate definition of R^2 should be provided, as well as its interpretation for survival data.

We thank the reviewer for pointing out a point of confusion in our description of the methodology. We have updated the Results section to describe how R^2 is used in evaluating fits to parametric forms. This involves a linear transformation of distributions followed by an evaluation of fit; in the case of Weibull

distributions this linearized form is a Weibull plot. This has the very convenient feature that R^2 then has the same interpretation as in linear regression (specifically, data that does not fit Weibull forms will not lie on a line in a Weibull plot, reducing R^2). We have revised Supplementary Data 2 to include both code and pseudocode (i.e. the method in plain English) to explain this computation fully.

3. P6, the second paragraph, given the observed R^2 is 0.981 and the theoretical maximum R^2 is 0.995, the interpretation that “Thus, the error between observation and the Weibull model is $\sim 1\%$ ” is incorrect. R^2 does not have such interpretation.

We apologize for sloppy language introduced during editing. The reviewer is of course entirely correct and we have changed this statement to “1-2% of variance is unexplained by Weibull model.”

4. P6, it is useful to show the boxplot of R^2 for 237 trial arms, including median, quantiles, in addition to the pooled mean.

We thank the reviewer for this suggestion and have included median, mean, and quantile values in the Results section (page 7, paragraph 2). We also now direct readers to a histogram of all computed R^2 values in Figure 2e and have added a boxplot to this subpanel.

5. The paper focuses on the marginal distribution of OS and PFS. It should be noted that the conditional distribution may not be Weibull. In practice, the main motivation of using nonparametric and semiparametric methods is to minimize assumptions and produce robust results.

We agree that the true underlying distribution of cancer patient survival data is unknown. The result of this study is not that survival distributions are Weibull distributions in an absolute mathematical sense but rather that empirical testing against a set of real survival data shows that a Weibull function describes the data with sufficient accuracy that for small trials it has nearly the same accuracy as nonparametric methods. Given this accuracy, use of parametric forms will increase precision, as we demonstrate.

We concur that nonparametric methods have been preferred historically because they avoid assumptions which, if incorrect, would compromise the robustness of results. This study shows that an assumption of a Weibull distribution is sufficiently close to reality to provide robust results. Making similar assumptions about form for a worthwhile gain in statistical power has strong precedent in clinical trial statistics; the most prominent example being the Proportional Hazards assumption of the Cox model.

We specifically see parametric methods as applicable to early-phase, signal-finding studies, where improving precision with acceptable accuracy could help greatly reduce the number of patients needed to understand the likely efficacy of a new therapy, saving trialists substantial time and resources. We do not suggest that this should replace established semi-parametric methods for pivotal trials and regulatory decisions. We have edited our Results and Discussion to better address these points (page 10, paragraph 3; page 16, paragraph 1). We note above other aspects of our study that do pertain to large Phase 3 trials and are made possible by using parametric fitting to analyze hitherto unavailable data.

6. The paper spent 3 pages to discuss the violation of proportional hazards. This fact is well known, with extensive publications, especially for ICI. I do not see the need of extensive discussion of this

under Weibull model, and is remotely related to the main result (PFS and OS can be well fitted by Weibull distribution). This section should be substantially shortened (<1 page).

We agree with the reviewer's call for brevity and have substantially shortened this section, with the revised text emphasizing the original contributions to this topic made possible only with large scale data and parametric fitting. We have also provided more context for prior contributions in this field through an updated Results section (page 13, paragraph 2).

This article's specific original contributions are (i) showing in diverse trials that proportional hazards is violated to some degree in a majority of clinical trials, far beyond ICIs, (ii) these violations can be understood as a consequence of differences in how the hazard increases or decreases over time (the shape parameter of the Weibull function); in this view it is understandable that different treatments would rarely have identical shape parameters, and thus rarely have proportional hazards, and (iii) as a consequence, trial length (follow-up time) impacts the likelihood of a positive outcome, entirely separately from how the number of events affects statistical power. This latter point is the most important and novel, and unfortunately topical because we see many Phase 3 trials failing that seem likely to have succeeded with longer follow up, such as KEYNOTE-022 (which had an initially negative result but a positive result in a later post-hoc analysis), and the recently published IMpassion131 trial (whose early censoring decreased power in the tail of the survival curve, resulting in the withdrawal of Atezolizumab for Triple Negative Breast Cancer).

Cancer patient survival can be accurately parameterized, improving trial precision and revealing time-dependent therapeutic effects ~~and doubling the precision of small trials~~

Authors: Deborah Plana^{1,2}, Geoffrey Fell³, Brian M. Alexander^{3,4}, Adam C. Palmer^{5*}, Peter K. Sorger^{1*}

Affiliations:

¹Laboratory of Systems Pharmacology and the Department of Systems Biology, Harvard Medical School, Boston, Massachusetts, USA.

²Harvard-MIT Division of Health Sciences and Technology, Harvard Medical School and MIT, Cambridge, Massachusetts, USA.

³Dana-Farber Cancer Institute, Boston, Massachusetts, USA.

⁴Foundation Medicine Inc., Cambridge, Massachusetts, USA.

⁵Department of Pharmacology, Computational Medicine Program, Lineberger Comprehensive Cancer Center, University of North Carolina at Chapel Hill, Chapel Hill, North Carolina, USA.

*These authors contributed equally. To whom correspondence should be addressed:

palmer@unc.edu; peter_sorger@hms.harvard.edu (cc:

sorger_admin@hms.harvard.edu~~sorger_admin@hms.harvard.edu~~)

ORCID:

Deborah Plana: 0000-0002-4218-1693

Geoffrey Fell: 0000-0001-6436-6691

Brian M. Alexander: 0000-0003-3903-9175

Adam C. Palmer: 0000-0001-5028-7028

Peter K. Sorger: 0000-0002-3364-1838

ABSTRACT

Individual participant data (IPD) from ~~completed~~ oncology clinical trials ~~are a valuable but rarely available~~ represent an invaluable source of information. ~~A lack of minable survival distributions has made it difficult to identify~~ for identifying factors ~~determining the~~ that influence trial success and failure ~~of~~, improving trial design and interpretation, and comparing pre-clinical trials and improve trial design studies to clinical outcomes. However, the IPD used to generate published survival curves are not generally available. We imputed survival IPD from ~500 arms of Phase ~~III~~ 3 oncology trials (representing ~220,000 events) and found that they are well fit by a two-parameter Weibull distribution. This ~~makes it possible to~~ finding supports the use of parametric statistics to substantially increase trial precision with small patient cohorts typical of Phase ~~I or II~~ trials. ~~For example, a 50-person trial parameterized using Weibull distributions is as precise as a 90-person trial evaluated using traditional statistics.~~ Mining IPD 1 and 2 trials. We also ~~showed~~ show that frequent violations of the proportional hazards assumption, particularly in Phase 3 trials of immune checkpoint inhibitors (ICIs),₂ arise from time-dependent therapeutic effects ~~and hazard ratios.~~ Thus, the. Trial duration of ICI trial therefore has an underappreciated impact on the likelihood of ~~their~~ success. All imputed IPD and analysis are available as supplementary materials and via the website <https://cancertrials.io/>

MAIN

INTRODUCTION

Extensive effort has been devoted to increasing rates of success in oncology drug ~~discovery~~ development by improving preclinical studies¹⁻³. However, completed ~~clinical~~ randomized

controlled trials (RCTs) remain the most valuable single source of information for understanding opportunities and challenges in ~~clinical~~ drug development. Retrospective comparison of ~~specific~~ trials is most commonly performed via meta-analyses and systematic reviews⁴ with the goal of improving patient management in specific disease areas⁵. Retrospective analysis has also been credited with improving the statistical treatment of trial data, which can be complex and confounded⁶. However, quantitative analysis of oncology ~~trial result~~trials is ~~very~~ difficult to perform at scale because individual participant data (~~IPD~~—namely times of progression, death and censoring; IPD), which are necessary for high-quality analysis, are rarely available^{7,8}. Trial results are ~~commonly~~instead reported in the form of summary statistics and, in the case of oncology trials, plots of patient survival based on the Kaplan–Meier estimator⁹. These plots are generated using IPD but it has proven time-consuming and resource-intensive to gain access to the underlying IPD values because journals and investigators do not generally make them available¹⁰.

To address this problem, the International Committee of Medical Journal Editors (ICMJE) recently ~~endorsed the concept of releasing IPD from clinical trials and~~ developed a set of data reporting standards to encourage release of IPD from all clinical trials¹¹. However, less than 1% of papers published in ~~the last three years actually comply with these standards~~ a two-year period were found to have made IPD publicly available¹². We and others have ~~therefore~~ developed methods to bypass this problem by using image processing to impute IPD values from published plots of the Kaplan–Meier estimator^{13–15}. In this manuscript we describe a comprehensive analysis of imputed IPD and reconstructed survival curves from ~150 publications reporting Phase 3 cancer trial results, which in aggregate comprise ~ 220,000 overall survival or event-free survival events (e.g. progression free survival, PFS). We also make these data freely available via an interactive website (<https://cancertrials.io/>). Our approach is consistent with the Institute of Medicine’s reports on best practices for sharing data from published clinical trials, including crediting the sources of the data and sharing all code used in the analyses¹⁶. ~~In this manuscript we describe a comprehensive analysis of~~

imputed IPD and reconstructed survival curves from 150 publications reporting phase III cancer trial results, which in aggregate comprise 220,000 overall survival or event-free survival events (e.g. progression-free survival, PFS).

We find that therapeutic responses as measured by overall survival (OS) or event-free survival fit well to unimodal distributions described by the two-parameter Weibull function; one parameter is proportional to median survival and the second quantifies changes in hazard over time. Analyzing survival functions with parametric distributions Analyzing survival functions with parametric forms of different types has a long history¹⁷, but evidence has been lacking about which distribution best represents real data. Parametric statistics are also well known to use; our analysis addresses this directly and shows increase precision, but only when the fit to data is sufficiently accurate. We now show that survival therapeutic responses for multiple cancer types and therapeutic classes as measured both by overall survival analysis using (OS) and event-free survival (e.g. progression free survival; PFS) are well fit by unimodal distributions described by the two-parameter Weibull forms increases precision without reducing accuracy. For example function; one parameter is proportional to median survival and the second quantifies changes in hazard over time. Using Weibull functions, we observe find that a 50-patient trial arm (assessing overall survival) is as accurate and precise as a 90-person trial arm evaluated using traditional nonparametric statistics; this finding is directly applicable to improving the precision of therapeutic efficacy estimates made with the small patient populations typical of Phase 1 and 2 oncology trials¹⁸ power in phase Ib/II trials. Weibull fitting of survival data also confirms that violations of the assumption of proportional hazards are common in contemporary Phase 3 trials¹⁵ notably for immune checkpoint inhibitors (ICIs)^{13,19} - but also more broadly. We find that differences in time-varying Violations arise from variation in hazard ratios between treatment arms cause over time and, as a consequence, so does the likelihood of trial duration to impact the likelihood of success. Simulation shows (which is most commonly defined as a hazard ratio <1 at a 95% confidence level). This effect is different from the increase in statistical confidence that occurs in any trial as a result of the accrual of

more events. In particular, simulation suggests that some failed trials with strong time-dependence might have been judged to be successful ~~(that is, to confer a hazard ratio <1 at 95% confidence)~~ had they been run for slightly longer. Trial characteristics computed from IPD ~~also make it possible to compare~~allow for comparison of response distributions across diseases and therapeutic modalities, potentially making it possible to improve the design of future trials, and reduce attrition, ~~and~~. The accuracy of the Weibull form in describing survival data may also assist cost-effectiveness ~~analyses by validating the assumptions necessary for such work in representative patient data~~research in which diverse parametric statistics are already in use^{22,20}.

RESULTS

Cancer patient survival ~~data~~ can be accurately parameterized ~~across many diseases and drug classes~~

We used previously described algorithms and approaches^{14,15} to mine published papers reporting the results of Phase ~~III research~~³ clinical trials in breast, colorectal, lung, and prostate cancer with endpoints including OS or surrogates such as PFS, disease-free survival (DFS), and locoregional recurrence (LRR) (which we henceforth consider in aggregate as “event-free survival”). ~~Briefly, image processing was used to extract~~For each trial between 2014 and 2016 that met our search criteria, plots of the Kaplan Meier estimator were extracted from trial figures using the DigitizeIt²², software (Braunschweig, Germany), while the at-risk tables and the number of patient events were manually extracted from the publication, ~~making it possible. We then used the digitized KM survival curves to impute estimate patient-level time-to-event outcomes (IPD (e.g., times of progression, death and censoring; Fig. 1A). Trial metadata were also curated and the resulting information is released in its entirety as supplementary materials to 1a).~~ We recently reported the use of this approach to reconstruct patient-level data for oncology Phase 3 clinical trial publications identified through a PubMed

search²¹ paper and via an online repository, (the accuracy and precision of IPD imputation is discussed in the **Methods**; see. Study-level information such as cancer type, metastatic status, treatment modality, and trial success was also **Supplementary Data Files S1-S2**), manually curated.

Analysis of OS data from 116 published figures (~~from 108 unique randomized controlled trial (RCT) reports~~) yielded 237 distributions (91,255 patient events) ~~and~~. Data on event-free survival from 146 figures (~~from 135 unique RCT reports~~) yielded 301 distributions (127,832 patient events). Classes of therapy included chemotherapy, immune checkpoint inhibitors, radiotherapy, surgery, targeted therapy, and placebo/observation. All imputed data were compared against the original trial publication for accuracy²¹ and trials with inaccuracies in the imputation procedure were excluded. The accuracy of IPD imputation is discussed further in **Supplementary Data 1** and **Methods**. The dataset is released in its entirety as supplementary materials to this paper and via an online repository, <https://cancertrials.io/>

~~Multiple~~ A variety of parametric forms have been proposed to describe survival in oncology trial data, including the Log Normal, Log Logistic, Gamma, Weibull, Gompertz–Makeham, and Exponential distributions^{23,24}. These differ in their hazard functions, which describe/quantify the likelihood of an event (e.g. death or progression) at a given time. ~~Specifically, the exponential distribution assumes a constant hazard, the Weibull, Gamma, and Gompertz–Makeham distributions allow hazards to change monotonically with time, and the Log Normal and Log Logistic functions allow for non-monotone hazard rates^{25–28}. To our knowledge, no systematic assessment of the accuracy of these forms in describing empirical data from a large set of oncology trials has previously been described. We therefore assessed the goodness-of-fit (R^2) of different parametric distributions to imputed IPD. First, “best-fit” parameter values were estimated for individual IPD distributions using a maximum-likelihood procedure (**Fig. 1b-c**). Second, mathematical transformations specific to a parametric form were used to linearize the distribution of event times and the corresponding survival values (in the case of the Weibull form the linearization is the Weibull plot)²⁹. These. Data that perfectly follow a proposed distribution would, in the transformed form, follow a straight line with $R^2 = 1$.~~

Parametric forms for survival distributions differ the most at long follow-up times when ~~their~~ “the tails” of the distributions fall to an asymptotic value or to zero. However, such long event times are rarely recorded in traditional oncology trials, which are limited in duration by cost and increased censoring (often because patients switch to an alternative therapy). As a consequence, we found that two types of two-parameter distributions fit survival data equally well: Weibull distributions and Log-Normal distributions (Weibull median $R^2 = 0.981$ and Log Normal median $R^2 = 0.980$). We ~~selected~~ chose to use the two-parameter Weibull distribution ~~for further analysis~~ because its parameters are easily interpreted in terms familiar to oncologists. The Weibull α (shape) parameter describes increasing or decreasing hazard over time³⁰, and the β (scaling) parameter is proportional to median survival time³¹. Survival data fit by Weibull distributions with having $\alpha < 1$ have decreasing hazard rates over time, meaning that the likelihood of progression or death is highest at the start of the trial and ~~decreases over time then falls~~. A value of $\alpha = 1$ corresponds to a constant hazard and $\alpha > 1$ to a hazard that increases with time.

For each trial arm in our data set, we ~~calculated the relative likelihood of different obtained best-fit~~ values of α and β ~~to describe its IPD (Fig. 1B-C) and found the best fit parameter values (Fig. 1D). It can be helpful to visualize the resulting 1d). The~~ distributions for individual arms and their parameterizations can be visualized in three different ways: (i) as ~~a~~ probability density function (PDF functions (PDFs)), the probability likelihood that an event will occur at any point in particular time t ; (ii) as ~~a~~ cumulative density function (CDF functions (CDFs)), the integral of the PDF with respect to t ; for OS data, 1-CDF is overall survival at t ; and (iii) as ~~a hazard function, which corresponds to the ratio of the PDF and survival function. A plot of patient level data as a PDF makes clear that death or progression is right skewed for all values of α observed here, so that a substantial proportion of all events occur well after the modal (peak) values (as illustrated in Fig. 1B). Fitting Weibull distributions therefore quantifies the frequently observed phenomenon that some patients’ response to therapy is substantially better than the most commonly observed responses to that treatment.~~ hazard functions,

which correspond to the ratio of the PDF and survival function. In oncology trials, the survival function is usually determined using the nonparametric Kaplan-Meier estimator (**Fig. 2a-c**), which accounts for progression or death events as well as censoring (e.g. ~~loss of follow-up within the trial duration or withdrawal from the trial for reasons other than progression or death~~). Hazard ratios (which measure the treatment effect relative to a control) and their confidence intervals are universally computed using Cox proportional hazards regression (referred to as Cox regression hereafter), a semi-parametric method. ~~withdrawal of a participant from the trial, or loss of follow-up, for reasons other than progression or death~~). A plot of patient-level data as a PDF shows that death or progression is right-skewed for all values of α that we observed in trial data (as illustrated in **Fig. 1b**). Thus, a substantial proportion of all events occur well after the modal (peak) survival value. Fitting Weibull distributions therefore quantifies the frequently observed phenomenon that the response of a subset of patients to therapy is substantially better than the most commonly observed response to that treatment.

For trials reporting OS data, we found that a two-parameter Weibull distribution had a median coefficient of determination of $R^2 = 0.981$ (lower quartile, Q1: 0.966, mean: 0.975, upper quartile, Q3: 0.989) across 237 trial arms from 116 figures in clinical trial reports (**Fig. 2d, 2e; Methods**); the histogram of R^2 values for every OS arm of every clinical trial can be found in Fig. 2e. The theoretical maximum R^2 value can be calculated under the hypothesis that all OS distributions are Weibull distributions and that deviations are attributable only to sample size variability, which gives yields a maximum $R^2 = 0.995$. Thus, the error between observation and ~2% of variance observed is not explained by the Weibull model is ~1%. For trials reporting event-free survival data (e.g. PFS; **Supplementary Fig. S1**) the 1 median R^2 was 0.950 (Q1 = 0.909, mean: 0.929, Q3= 0.970) as compared to a theoretical maximal $R^2 = 0.996$, which corresponds to 5% of observed variance not explained by the Weibull model. Biomarker-stratified arms were also well described by a single two-parameter Weibull distribution. This is illustrated in Supplementary Fig. 2a-b for Weibull fits to OS and PFS data for panitumumab in combination with FOLFIRI and for FOLFIRI alone in wild-type and

mutant *KRAS* metastatic colorectal cancer (trial 20050181³³; average $R^2 = 0.99$ for OS curves and 0.92 for PFS curves). We conclude that a two-parameter Weibull distribution provides an excellent fit, with the few exceptions discussed in detail below, to available trial data across multiple types of cancer, treatment modalities, and metastatic status.

Investigating the least good fits of survival data to Weibull parametrizations

Across the entire data set, some of the worst fits for two parameter Weibull forms were observed for trials with relatively few events, for example, the Chronicle trial (NCT00427713³⁴) with only 78 deaths in the treatment arm and 16 progression events in the observation arm (**Fig. 2c; Supplementary Fig. S11**). Fit was also poor for trials involving pre-planned changes in treatment such as the ACT2 trial (NCT01229813)³⁵, in which treatment induction was followed by randomization to maintenance treatment at 18 weeks (**Fig. 2c**). In these cases such as this, responses varied over the course of the trial by design and a good fit to a single two-parameter survival function was not expected.

For two a small number of trials in which asymptote of the lowest quality fits, fitted survival curve was greater than zero (i.e. patients were expected to be alive at the end of the longest follow up), a three-parameter Weibull distribution (consisting of the traditional two-parameter distribution with an additional “cure rate” term³⁶) resulted in had an improvement in the quality of improved fit (Supplementary Fig. S2). However, 3. Since the improvement in fit was modest and two-parameter Weibull forms had excellent performance for the bulk of the data, are generally both more computationally tractable for and are more parsimonious, we relied on them for all subsequent large-scale analysis, and are more parsimonious. However, the use of a cure rate parameter might nonetheless be advisable for different sets of data in which cure is a known outcome (e.g. R-CHOP for non-Hodgkin’s lymphoma³⁷). Biomarker-stratified arms were also described well by a single two-parameter Weibull distribution, as illustrated by the OS and PFS curve fits for panitumumab in combination with

~~FOLFIRI and FOLFIRI alone in wild-type and mutant *KRAS* metastatic colorectal cancer (trial 20050181; Supplementary Fig. S3A-B). We conclude that a two-parameter Weibull distribution provides an excellent fit to available trial data across multiple types of cancer, treatment modalities, and metastatic status).~~

~~Deviation from the one-distribution Weibull model in PFS data~~

~~The~~ We also observed that event-free survival exhibited a slightly poorer fit ~~of event-free survival data to Weibull distributions as compared to~~ OS data (5% vs 2% of observed variance not explained). Inspection of the relevant curves showed that ~~this~~ was caused primarily by a ~~steep fall seen~~ sharp decrease in ~~some~~ survival curves at early time points ~~and a shallowed decrease subsequently~~; this behavior has previously been interpreted as evidence for subpopulations of responding and non-responding patients, particularly in trials of ~~immune-checkpoint inhibitors (ICIs³⁸)~~. It has also been attributed to delayed T-cell activation by ICI therapy³⁹. For trials of these agents, we found that ~~fit~~ fits to PFS data could be improved by using a mixture model comprising two different two-parameter Weibull distributions, each with its own α and β parameters. This is potentially consistent with ~~the~~ two-population hypothesis (Fig. 3a; Supplementary Fig. S44). However, ~~the same was a mixture model also true of~~ resulted in a better fit to the control arms in these trials, suggesting that the deviation from a single distribution ~~at early time points~~ was not ICI-specific. Moreover, a mixture model exhibited no ~~meaningful~~ improvement in fit as compared to a single-distribution ~~fit~~ for OS data from ICI or control arms ~~in these trials~~ (Fig. 3b; Supplementary Fig. S4-4). Finally, when we examined PFS data from an additional 25 ICI trials, we found that the drop in ~~patient~~ survival at early event times (~~which we identified as determined by finding~~ the time t corresponding to the greatest change in the slope of the survival curve) was strongly correlated with the time of the first radiological scan (as reported in trials' methods sections; Pearson correlation 0.982, $p < 10^{-21}$; Supplementary Data File S33). We ~~then surmised that fitting Weibull distributions to PFS data was confounded by scan times. To test this~~

~~idea we~~ simulated the influence of scan times ~~on PFS~~ by ~~beginning with taking~~ a single ~~2-parameter~~ Weibull distribution; and ~~supposing that progression could only be observed when imposing~~ a tumor is scanned, which occurs periodically (often at ~~scanning interval of 9 or 12-week intervals depending~~ ~~on weeks~~ (the actual value was extracted from the trial protocol). ~~Accounting for scan time recapitulated~~ ~~the initial~~ This generated the steep decline in PFS ~~values observable~~ in ~~both the~~ control and experimental arms of ~~actual~~ ICI trials, and improved fit to PFS distributions, raising mean R^2 ~~from to 0.93~~ to 0.98; ~~compared to 0.92 without considering scan times~~ (Fig. 3c; Supplementary Fig. S44; Methods). We conclude that ~~the initial a steep~~ drop in ~~initial~~ PFS arises when observations corresponding to the ~~left-hand tail of~~ is likely to arise because values at early time points from a unimodal response distribution are concentrated in time ~~because radiological by~~ scans ~~used to measure tumor progression are~~ performed at discrete intervals. ~~Thus, We further conclude that~~ mixture models involving two Weibull curves ~~do are~~ not ~~appear~~ necessary to accurately describe survival for ICIs or any other class of therapy that we have examined. Instead, when scan times are accounted for, single two-parameter Weibull distributions are ~~observed found~~ to have an ~~outstanding excellent~~ fit ($R^2=0.98$) to PFS data. ($R^2=0.98$).

Parametric fitting improves the precision of drug efficacy estimates

To compare the performance of Weibull-based ~~methods~~ and nonparametric ~~methods used for~~ survival ~~estimates~~,

~~analysis~~ we calculated ~~12-month~~ pointwise confidence intervals (at 12-months). This is a frequently reported statistic for many early phase oncology trials (Phase 1 and 2) that involve relatively small numbers of patients. It is often referred to as a “landmark” outcome in systematic reviews and meta-analysis of oncology clinical trials⁴⁰, and used to guide the design of larger trials. A challenge in the analysis of such data is that, when too few events have occurred, nonparametric numeric confidence estimates return non-informative values (usually reported as a value “not reached” or “indeterminate”²⁸ for “small cohorts” simulated by sampling as illustrated for two different scenarios in

Supplementary Fig. 5a). To determine how parametric analysis would perform in this setting, we subsampled groups of 20 to 100 patients at random from data~~the arms of~~ imputed from real trial arms. Nonparametric methods are unable to report a numeric confidence interval when too few events occur before or after the time point at which the analysis is being conducted (12 months in our simulations; “failed” Phase 3 trials. We then compared estimates correspond to 100% or 0% of 12-month survival respectively). Infor small cohorts this issue is with a well-known to limit the power of nonparametric analysis. powered ground truth value obtained using the full Phase 3 dataset. We found that ~~no~~ numerical confidence interval could be computed 12-month survival estimates were non-informative for 20% to 40% of OS trial arms, and ~~for~~ 23% to 61% of event-free survival trial arms, with higher failure rates occurring with when the sample size was smaller ~~simulation sample sizes~~. (Supplementary Fig. 5b-c). In comparison, use of a Weibull parametric form made it possible to calculate 12-month confidence intervals in every case for all of the ~45,000 simulated 20 to 100 patient trials that we examined (~~~19,000 OS and ~25,000 event-free survival simulations~~). The same advantage applies to median survival time, which is relevant because many early phase oncology trials report median survival (or EFS, PFS) with an estimated upper bound that is “not reached” and therefore uninformative. Thus, Weibull fitting is broadly applicable in survival analysis of small cohorts because it makes it possible to reliably obtain confidence intervals for median survival and for time points of interest.

~~For~~ Considering only the subset of OS curves ~~infor~~ which confidence intervals could be computed using both parametric and nonparametric methods (125 curves), ~~the~~ we found that a Weibull-based approach was more precise (it had a narrower confidence interval) across all sample sizes, and accuracy was comparable. By way of illustration, the precision of a 50-patient trial was comparable to that of a 90-patient study using traditional methods (**Fig. 4**). For event-free survival (for which 99 event-free survival curves could be compared), the precision of a Weibull-based approach was also greater than a nonparametric approach across all sample sizes (Supplementary Fig. S5A). Thus, modeling cancer patient survival using Weibull functions increases precision, without compromising accuracy. The impact is greatest for trials having

sample sizes typical of phase Ib or II trials. This is also a setting in which use of innovative statistical methods is most likely to be acceptable from a regulatory standpoint, while accuracy was equivalent for small sample sizes (fewer than 40 patients) (Supplementary Fig. 6a). We conclude that the use of Weibull distributions to parameterize data from small trials reporting either OS or PFS data approximately doubles the number of trials for which informative confidence intervals can be determined for a point survival estimate (e.g. survival at 12 months). Moreover, for the subset of trials in which parametric and nonparametric methods can be directly compared, the former is as precise using roughly half the number of patients.

Figure 4 indicates that survival estimates made using Weibull parameterization decrease in relative accuracy as compared to nonparametric methods as patient number is increased. This arises simply because nonparametric analysis of the full set of Phase 3 data was defined as the ground truth. It is nonetheless true that using Weibull forms is most valuable when cohorts are small (fewer than ~40 patients). This sample size is typical of Phase 1 or 2 oncology trials, a setting in which alternative statistical methods are also most likely to be acceptable from a regulatory standpoint.

As one illustration of the use of Weibull parameterization, we analyzed a recent trial that encompassed both Phase H1 and H2 data, and tested pembrolizumab with dabrafenib and trametinib for metastatic *BRAF*-mutant melanoma (MK-3475-022/KEYNOTE-022; NCT02130466)⁴¹. Parametric fitting for 15 patients in Phase H1 yielded a median ~~value~~PFS of ~~15.2~~14.8 months and 95% confidence interval ~~for median PFS~~ of 7.8 to 23 months, while nonparametric estimates yielded a median value of 15.4 months and 95% confidence interval of 5.4 months to “not reached” ~~(median value 15.4 months)~~. Nonparametric analysis of a Phase H2 cohort of 60 patients ~~within~~for this same trial ~~demonstrated~~revealed a median PFS of 16 months and a 95% confidence interval of 8.6–21.5 (Ref⁴²). Thus, parametric fitting of data from 15 patients made a comparably precise and accurate estimate of median PFS as nonparametric analysis of 60 patients (Supplementary Fig. S5B6b). The availability of a more precise inferences parametric approach would ~~make~~in principle have made it possible to use the same number of patients enrolled in ~~single phase H~~this Phase 2 study (e.g.: 60 patients) to perform three

different signal-finding studies (each involving 20 patients) with no loss of statistical power. This might~~would~~ have been particularly helpful in the case of KEYNOTE022, a trial ~~which~~that failed to meet its primary endpoint.

Evaluating the impact of patient heterogeneity on the accuracy of Weibull parameterization

Subsampling Phase 3 trials to generate synthetic arms having the small numbers of patients typical of Phase 1 and 2 trial cohorts has the advantage that the Phase 3 data serve as the ground truth. However, it has the disadvantage that patient populations in early stage trials are often more heterogenous than in pivotal trials. We have been unable to identify a sufficient number of matched early and late phase survival data for comprehensive investigation of this issue. As an alternative approach, we simulated a trial having a heterogeneous population of ~900 patients with a mixture of breast, colorectal, lung, and prostate cancer cases. The simulated cohort was constructed by subsampling five patients from each of 172 trials arms that reported OS for patients with metastatic cancer. We observed that for a representative simulation, a two-parameter Weibull form accurately described the synthetic trial data ($R^2=0.98$; **Supplementary Figure 7**). However, a representative synthetic cohort involving patients drawn from both metastatic and local cancers (237 trial arms) is not as well fit by a two-parameter model ($R^2=0.95$) but fit improves with the addition of a cure-rate parameter ($R^2=0.98$). From this analysis we conclude that trial arms having different types of solid tumors, as encountered in some basket trials, can be accurately parameterized by two-parameter Weibull functions. If metastatic and non-metastatic disease are mixed in an RCT, addition of a third parameter is likely to improve fit.

Weibull fitting quantifies survival differences ~~in survival~~ across cancer types

The availability of a large set of IPD made it possible to search for systematic differences in the parameters of survival distributions by disease class. Best-fit Weibull parameters were compared across cancer types and metastatic status using an ANOVA test with a Bonferroni correction for multiple

hypothesis testing at a two-tailed significance level of 0.05 (see **Supplementary Data File S22**). The largest difference in parameter difference values was between metastatic and non-metastatic disease, irrespective of tumor type (β values corresponding to median survival of 2622 and 174180 months respectively). We also observed that β values were significantly larger for breast cancer than lung cancer on a variety of experimental and control treatments in the metastatic setting (β values corresponding to median survival of 3828 versus 1514 months in the metastatic setting) (**Fig. 5**), which is consistent with previous data on relative disease severity^{43,44}. Parameter values for trials reporting event-free survival parameter values were followed a similar pattern to OS values (**Supplementary Fig. S68, Supplementary Data File S22**). Lung cancers had a significantly lower α (shape parameter) for OS as compared to other cancer types (average $\alpha = 1.30$ for lung; versus ~ 1.5 to 1.6 for breast, colorectal, and prostate cancers), demonstrating a relatively high probability of early death in the course of treatment, even when controlling for differences in median survival times. This difference in shape corresponds to a widerwide distribution of lung cancer survival times as compared to other cancer types (e.g.: the ESPATUE, which may reflect heterogeneity in lung cancer trial had a survival time distribution with a 10th percentile value of 3 months and a 90th percentile value of 49 months; Fig. 5C), cohorts. We propose that parameters drawn from IPD could be used to model cancer survival distributions across diseases, facilitating for future exploratory trials and to facilitate inter-group comparison comparisons in master protocol or basket trials, which often involve different cancers types^{44,45,46}.

Violations of proportional hazards and Impact of trial length on estimates of relative hazard

Randomized controlled trials in oncology cancer are conventionally evaluated in a majority of cases based on hazard ratios; the use of Cox regression to estimate the semi-parametric hazard ratio (hereafter referred to as HR_{SP}). If the hazard ratio is significantly below one then the test treatment has decreased decreases the risk of death or progression relative to control, and the trial is regarded as

successful^{28,32}. ~~Cox regression estimates the semi-parametric hazard ratio (hereafter referred to as HR_{SP}) based on the number and timing of death, progression, or censoring events, all of which increase over time.~~ As expected, when Weibull α and β parameter values were compared between experimental and control arms, a trial was more likely to be successful (~~HR_{SP} which, following common practice, we defined as HR_{SP} < 1 at 95% confidence~~) when differences in β values were ~~greater~~larger: the median difference between control and experimental β values in OS curves was ~~-0.946%~~ for unsuccessful trials and 29% for successful trials (**Fig. 6A**). A similar pattern was observed for event-free survival data, with control and experimental β values differing by 1.0% for unsuccessful trials and ~~35.736%~~ for successful trials (**Supplementary Fig. S79**).

Fundamental to the model of proportional hazards is the idea that the hazard functions for control and experimental arms are ~~described by hazard functions~~ related by a constant of proportionality (the hazard ratio) that does not change over time. However, prior work has shown that this assumption is frequently violated^{13,15,47}. From the perspective of Weibull distributions, ~~this proportionality~~ means that the two arms have the same shape parameter (i.e. $\Delta\alpha = 0$ where $\Delta\alpha$ is the difference in α values, ~~$\Delta\alpha$ is zero~~); ~~the~~. A trial ~~will be~~with $\Delta\alpha = 0$ is successful (~~hazard ratio < 1~~) if the experimental arm has significantly larger β value than the control arm. However, across 121 comparisons of experimental and control arms from 116 OS trial figures, we found that $\Delta\alpha$ values actually varied from +0.65 to -0.80 (median absolute value $|\Delta\alpha| = 0.11$; **Fig. 6a**). For event-free survival data, ~~154~~155 comparisons of experimental and control arms from 146 trial figures revealed a range of $\Delta\alpha$ values from +0.55 to -0.85 (median $|\Delta\alpha| = 0.08$) (**Supplementary Fig. 9**). Using a traditional Grambsch–Therneau test⁴⁸ ~~to -0.85~~ (median $|\Delta\alpha| = 0.08$) (**Supplementary Fig. S7**). Thus, ~~the assumption in the proportional hazards model that $\Delta\alpha = 0$ is frequently violated. To compare this to previous reports of violation of proportional hazards, we used a Grambsch–Therneau test at a significance level of $p < 0.1$. In our dataset, 18 out of 108 unique publications reporting OS data (~17%) were found to violate proportional hazards, and a significant deviation from the proportional hazards assumption corresponded to $|\Delta\alpha| = 0.30$. For unique~~

trials reporting event-free survival data, 47/135 (~35%) were found to violate proportional hazards by the Grambsch-Therneau test at 10% significance, and this significance cutoff also corresponded to $|\Delta\alpha|=0.30$. ICI trials tended to violate proportional hazards at a higher rate compared to all other trials (4/5 trials). Thus, differences in Weibull α parameters greater than 0.3 for control and experimental trial arms identify significant deviations from the assumption of proportional hazards and are most common in ICI trials. a proportional hazards violation was found in 18/108 OS and 47/135 event-free survival trials, and 8/10 ICI trial comparisons (3/5 for OS and 5/5 for PFS)¹⁵; a corresponding 90% confidence level yields a significance threshold of $|\Delta\alpha|=0.30$.

~~We used parametric fitting~~ To explore the originorigins and consequenceconsequences of non-proportional-proportionality of hazards in trialsurvival data. Specifically, we used best-fit Weibull shape and scale parameters for each trial arm (and confidence intervals thereof) to calculate the ratio of cumulative hazards between experimental and control arms hazard at time t ($HR_c(t)$), for all values of t from the start to the end of the trial. This approach returned robust estimates of relative empiric hazard for each trial arm as a function of an approach that makes no assumptions about proportionality, and compared $HR_c(t)$ to the hazard ratio calculated using Cox regression (HR_{SP} ; which is semiparametric and time without assuming proportionality. Trials that deviated little from the proportional hazards assumption, such as the failed ATTENTION trial (NCT01377376) with $\Delta\alpha = 0.003$ or the invariant). In both successful RECOURSE Trial (NCT01607957) with $\Delta\alpha = 0.13$, Weibull and unsuccessful trials for which $|\Delta\alpha|$ was small, $HR_c(t)$ (blue lines in Fig. 6b, c) closely approximated HR_{SP} from Cox regression ((red lines; note that the HR_{SP} under the assumption of proportional hazards is, by definition, time-independent). In the case of). In contrast, when $\Delta\alpha = -0.26$ (for the successful trial CheckMate 057 (NCT01673867⁴⁹) a value $\Delta\alpha = -0.26$ denotes a deviation from proportional hazards and we observed that $HR_c(t)$ matched and HR_{SP} at only one point in time differed for most of the trial duration (Fig. 6E6d). This was also true of CA184-043 (NCT00861614) for which $\Delta\alpha = -0.30$; this trial), which was judged to have failed based on HR_{SP} ⁵⁰ (Fig. 6D)6e). However, in this trial, the $HR_c(t)$ also matched HR_{SP} from

~~Cox at a single point in time but it~~ fell steadily ~~to <1~~ over time and had reached a value below one by the end of the trial. Unless the shape of the hazard function were to change substantially ~~change over one additional~~ after month 25, it seems probable that CA184-043 would have been judged a success ~~by conventional HR_{SP} criteria had had~~ it continued slightly for only a few months longer.

To more fully explore the dependence of trial duration on outcome, we simulated a series of two-arm trials having a range of differences in $\Delta\alpha$ and $\Delta\beta$ values. We then plotted the fraction of trials that were successful, as judged by Cox regression. Success was evaluated both at an early stopping point, when ~60% of events had been recorded (t_A) or a late stopping point when ~95% of events had been recorded (t_B ; Fig. 7; Methods). For simulated trials in which α was smaller for the experimental than the control arm, the experimental arm exhibited lower survival at early times and then crossed over the control arm at later times (as shown in Fig. 7a). In these cases, a later time point was associated with a greater likelihood of success than an earlier time point (Fig. 7b,c). The greater the value of $\Delta\alpha$, the greater the impact of curve crossing and duration of follow-up on outcome. Moreover, all ICI trials in our data set fell into this category (e.g. Fig. 6a). The reasons for time-dependent therapeutic effects are unknown, but in ICI trials it has been suggested that they arise from treatment-related toxicity at early times or delayed treatment effects^{13,52,53}.

The importance of trial duration on success was recently demonstrated ~~in~~by the MK-3475-022/KEYNOTE-022 trial of pembrolizumab with dabrafenib and trametinib for *BRAF*-mutant melanoma, ~~where~~. The pre-planned analysis at 24 months did not identify a statistically significant benefit (PFS HR of 0.66, 95% CI: 0.40-1.07) but a subsequent analysis at a median 36.6 months ~~of follow-up~~ did (PFS HR of 0.53, 95% CI: 0.34-0.83)^{42,51}. ~~We conclude that violations of the assumption of proportional hazards in clinical trial data do not simply involve statistical deviations, but actual variations in treatment effect over time—and this is true of both OS and PFS data. This contrasts the assumption in Cox regression that HR_{SP} has a fixed value over the course of the trial, such that time~~

enters into consideration only insofar as enough events must accrue for HR_{SP} to be judged significantly different from one. Parametric fitting of the original Phase 2 data at 24 months found a difference in the survival curve α values ($\Delta\alpha=-0.21$). This is the scenario in which a statistically significant benefit from therapy is more likely to be identified at longer follow-up, as was confirmed by data at 36.6 months.

To more fully explore the dependence of trial duration on outcome, we simulated a series of trials in which control and experimental arms exhibited a range of differences in α and β values and then plotted the fraction of trials that were successful, as judged by Cox regression under a proportional hazards assumption ($HR_{SP}<1$ at 95% confidence). In evaluating “success”, we applied Cox regression and ignored violations of proportional hazards as is the currently accepted standard (see Discussion). Success was evaluated at a stopping point defined by 60% of events being recorded (t_A) or a late stopping point of 95% of events being recorded (t_B ; **Fig. 7; Methods**). For simulated trials in which α was smaller for the experimental than the control arm, the likelihood of success was substantially greater at the late stopping point as compared to the earlier stopping point (**Fig. 7A-B**; conversely, early termination, for apparent futility for example, would incorrectly support a conclusion of inferiority). In these cases, we see that the experimental arm exhibited lower survival at early times and then crossed the control arm at a later time to exhibit higher survival. From these data we conclude that oncology trials exhibit continuous deviations from the assumptions of the proportional hazards model. The underlying variation in treatment effect over time can be identified by Weibull fitting as situations in which $|\Delta\alpha| \gg 0$. In these cases, the duration of the trial can have an effect on the likelihood of success in a manner that is not accounted for by Cox regression. We suggest that future trials, particularly of ICIs, evaluate $\Delta\alpha$ and model the possible impact of trial duration on the likelihood of success.

~~The greater the value of $\Delta\alpha$, the greater the impact of curve crossing and duration of follow-up on outcome.~~ All ICI trials in our data set fall in this category, since their experimental arms have smaller α parameters than their control arms (**Fig. 6A**), resulting in decreasing $HR_e(t)$ over time and curve

~~crossing. Thus, the time at which these trials are terminated impacts outcome, independent of the number of events needed to reach statistical significance. The reasons for time-dependent therapeutic effects are unknown, but in ICI trials it has been suggested that they are related to initial treatment-related toxicity or delays in treatment effect. We conclude that future ICI trials should consider the impact of time-varying changes in relative hazard (which are detectable via Weibull fitting) on trial success as a factor independent of the number of events needed to reach statistical significance in the estimate of HR_{SP}.~~

DISCUSSION

Using a set of ~220,000 imputed participant survival events from published oncology trials we find that survival functions for solid tumors, including those from trials that report OS or event-free survival data (i.e.: PFS), or are biomarker-stratified, are well fit by two-parameter Weibull distributions. The poorest fits are often explainable by pre-planned changes in treatment and by the confounding effects of radiological scan times on evaluation of PFS. The Weibull α (or shape) parameter defines increasing or decreasing hazard over time and the β parameter is proportional to the median survival time, making fitted parameter values readily interpretable. Both α and β differ between treatment and control arms; $\Delta\alpha$ quantifies violations in the assumption of proportional hazards that is used in Cox regression and $\Delta\beta$ measures the magnitude of the therapeutic effect. The excellent fit of survival data to a single parametric function for many types and stages of cancer and across drug classes demonstrates that therapeutic benefit can be well-described by a simple function ~~that models response as varying in~~ which responses vary continuously across a population. In the trials studied here, the likely presence of prognostic factors, or responder and non-responder populations, did not sufficiently separate survival functions to produce bimodal distributions that would have necessitated the systematic use of mixture models or cure-rate parameters.

Our findings support modeling survival in early stage oncology clinical trials by using parametric statistics. Parametric statistics are already used in cost-effectiveness analysis²⁰ and other simulation studies. ~~Our, although it has not been established which parametric forms (Weibull, Log Normal, Gompertz–Makeham, etc.) accurately fit empirical survival data uses a large and diverse clinical data set to verify. We now establish~~ that Weibull distributions ~~are the preferred parametric form and that they~~ involving clinically interpretable parameter values have sufficient accuracy to be the preferable parametric form for applications describing survival in trials of solid tumors. By simulating trials of different sizes, we find that modeling with parametric statistics substantially improves precision with equivalent accuracy: ~~assessing 12-month point estimates of survival outcomes (e.g. at 12-months) using a Weibull-based parametric approach makes approximately doubles the number of trials in which informative confidence intervals can be obtained. In cases in which parametric and nonparametric approaches can be compared directly, we find that~~ a 50-person trial reporting OS data is as precise as a 90-person trial evaluated using nonparametric methods. ~~This underestimates the actual benefit of using Weibull forms because nonparametric methods did not return a numerical confidence interval in 20–40% of OS simulated~~ This advantage pertains primarily to trials with small numbers of participants (20–100 patients, whereas parametric estimation was successful in all cases per arm); when arms are larger, the advantage of parametric statistics disappears and conventional Cox regression is the preferred method. Thus, ~~the~~ use of parametric statistics based on Weibull forms should be strongly considered in early phase signal-seeking studies with the goal of rapidly and economically identifying the optimal setting in which to perform Phase ~~III~~3 trials.

Weibull distributions are also appropriate for cost-effectiveness research for oncology drugs, an increasingly important topic for drug approval in many countries. In this context, it is important to note that Weibull and Log-Normal distributions provide equivalently good fits to IPD and the Weibull form was chosen in the current study because of its interpretability in terms of hazard rates and median survival. Log-Normal distributions may have corresponding advantages in pharmaco-economic

analysis⁵⁴. Moreover, insofar as there exist multiple ways to implement parametric statistics, we note that our results pertain specifically to approaches detailed in the **Methods** (~~these, which are largely conventional~~). Alternative ~~statistical~~ approaches to increasing trial precision, for example by changing significance ~~cut-offs of traditional confidence intervals (levels~~ to create narrower nonparametric confidence intervals while still maintaining an acceptable level of Type I error), have not yet been empirically explored in detail but can be pursued using the imputed IPD provided with this manuscript.

In current practice, Cox regression is used to compare survival functions based on the proportional hazards assumption, which states that the ratio of control and experimental hazard ~~functions~~rates is constant over time. Success usually corresponds to $HR < 1$ at 95% confidence. With respect to Weibull distributions, the assumption of proportional hazards corresponds to no difference in shape, i.e. $\Delta\alpha = 0$. It is well established that a subset of trials deviate from the proportional hazards assumption^{13,15}. However, ~~in real trial OS data, we find that $\Delta\alpha$ was found to vary~~varies over a wide range, from 0.65 to $\sim +0.80$ with ICI trials as a class having the largest $|\Delta\alpha|$ values.7 to -0.8, and that the majority of trials analyzed deviate from the proportional hazards assumption to some degree. If we use~~apply~~ previously described criteria (the Grambsch–Therneau test at 10% threshold)⁴⁸ ~~to determine which violations of proportional hazards are to identify~~ significant deviations from proportional hazards^{13,15} we find ~~that they correspond to $\Delta\alpha > 0.3$ and apply to violations in~~ $\sim 17\%$ of trials reporting OS and $\sim 35\%$ of trials reporting PFS data, and that this significance level corresponds to $|\Delta\alpha| > 0.3$.

Analysis of imputed data from published trials, and simulations ~~in which~~using empirical survival functions ~~are resampled~~, shows that violations of proportional hazards ~~are not statistical curiosities but instead~~ arise from time-varying treatment effects. In the data analyzed here, this was most evident in ICI trials, but was also seen in trials of the BCL-2 inhibitor venetoclax, in which experimental and control ~~arms~~arm PFS curves cross each other ~~some time~~ after trial initiation⁵⁵.

The biological basis of time-varying treatment effects (and curve crossing) are not known ~~in detail~~ but could arise from high toxicity in a subset of patients early in treatment, delayed onset of

treatment effects, or exceptionally durable responses in some patients (indicating the presence of prognostic factors)^{13,15,52,53}. Regardless, the practical consequence of these effects is that the duration of a trial has a direct impact on outcome, independent of statistical considerations such as increasing confidence in HR values as trial events accrue (as in Cox regression).

We found that it was possible to use Weibull fitting to identify trials judged as failures by Cox regression in which HR was trending steadily below one at the end of a trial ~~but, and~~ an extension of only a few months was predicted to result in success. ~~We suggest that such considerations be taken into account in~~ Future work could explore the design of future trials Weibull fitting in trial interim analysis, particularly in trials where $|\Delta\alpha| \gg 0$ such as for ICIs.

~~Additional information that can be mined from IPD, to improve future help determine when to terminate trials includes for treatment futility. Additional changes that could be implemented in future trials include improving~~ how sample size and power are estimated. Such calculations are most commonly performed under an assumption of an exponential fit to survival data. Alternative distributions have been proposed for such analyses⁵⁶⁻⁵⁹ but without any means for selecting optimal parameter values for simulation. Using the Weibull fitting described here, empirically-derived parameter values can be drawn directly from past trial data. ~~In some cases, past trial data might also be useful for the generation~~ A final set of synthetic control arms. ~~Additional applications include involves~~ the use of parametric forms for subgroup analysis in Phase III and basket trials. Since studies of this type are intended to test therapeutic hypotheses rather than lead to drug registration, the regulatory barriers to using parametric statistics are limited. Parameterized Bayesian trial designs, such as the continual reassessment method (CRM) or escalation with overdose control (EWOC), are other model-based methods already in use ~~used~~ to define specific parameter values and improve the efficiency of Phase I studies⁶⁰.

Even modest improvements in the design and interpretation of oncology clinical trials are likely to have a substantial payoff. The overall approval rate for new oncology drugs remains low: only 3% of

drugs tested in a Phase H1 clinical trial and 7% of drugs tested in a Phase H2 study are ultimately found to be superior to standard of care comparators in pivotal Phase H3 studies⁶¹. Methods to more accurately understand drug activity in small patient populations are included in the National Cancer Institute's (NCI) 2020 “provocative questions” and could lead to a wider use of master protocol trials. Improving trial efficiency and predictability will become increasingly critical as the number of new monotherapies and combinations continues to rise, patient populations become more subdivided based on the molecular characteristics of their tumors, and it becomes impractical to enroll enough patients to test all promising drug treatments⁶².

The use of nonparametric statistics was historically appropriate because treatment effects could be calculated precisely without the need for extensive computation³², which was largely infeasible prior to the widespread availability of personal computers. Moreover, the proportional hazards assumption appears to be largely valid when scoring/assessing OS in the context of cytotoxic chemotherapies (OS trials including chemotherapies in our data set had a median $|\Delta\alpha| = 0.11, 10, \text{ well}$ below the $|\Delta\alpha| = 0.30$ threshold for significant violation). However, the ~~widespread violation of deviation from~~ proportional hazards reported in this and previous studies, and its likely origins in the biology of new and more diverse forms of cancer therapy, call for a reconsideration of Cox regression. Several approaches for comparing treatment effects have been proposed including weighted⁶³ or adaptive log-rank tests⁶⁴, restricted mean survival test times^{65,66}, and permutation-based approaches⁶⁷. ~~It is probably appropriate for. Trialists, sponsors, and~~ regulatory agencies may want to examine the ~~possible~~ use of such methods in the setting of the ICI trials that are the focus of so much current research.

Limitations of this study

This work is not a formal meta-analysis or systematic review of a specific treatment regimen or disease, but instead a broadly conceived research study; no treatment decisions should be made based on

our findings. ~~A specific limitation of this study is that it uses imputed IPD rather than original data. A related~~A specific limitation is that we only analyze four tumor types (breast, colorectal, lung, and prostate); extending the analysis to ~~additional~~other cancer types will require imputing IPD from additional trials. ~~Finally~~Additionally, we use Cox regression to determine whether real or simulated trials are “successful” (e.g. **Fig. 6 and 7**) even when ~~underlying~~their survival distributions clearly violate proportional hazards. We do this because a finding of $HR < 1$ at a pre-specified level of confidence is the only widely accepted method for evaluating trial outcomes. ~~Potential limitations in parametric approaches may be addressable by reconstructing a larger number of trials for additional cancer types; however, this is a substantial undertaking.~~Finally, we used subsampled patient events from completed, Phase 3 trials to infer the properties of small patient populations commonly used in Phase 1 and 2 trials on the assumption that underlying survival functions have similar parametric forms in early and late stage trials. We are unable to rigorously assess this assumption but we find the simulated trials comprising four different cancer types are also well fit by two-parameter Weibull forms suggesting that having a heterogenous patient population (as encountered in many early phase trials) does not reduce the accuracy of parametric analysis. The parameters of best-fit Weibull distribution are very likely to differ between Phase 1 or 2 and Phase 3 studies whenever there are differences in inclusion criteria, such as prior therapy, performance status, tumor stage, and histology. Notably, a similar limitation can be expected from other methods (e.g. traditional nonparametric approaches) that use early-phase efficacy data to design pivotal studies.

A final limitation of this study is that it uses imputed IPD rather than original data. We are forced to ~~use imputed IPD~~do this because original results are simply not released, and most published oncology trial reports do not provide the numerical values used to plot the Kaplan-Meier estimators. Release of numerical data underlying graphical representations has become the norm in pre-clinical research and is at the heart of efforts by funding agencies to make data FAIR (findable, accessible, interoperable and reusable). Multiple calls have been made to make IPD from research clinical trials publicly accessible

to ensure the reproducibility of study results and facilitate meta-analyses, but compliance remains low¹². Outside of oncology, calls for reuse of both contemporary and historical control arms have arisen in repurposing trials for COVID-19, particularly when the same set of institutions is conducting many parallel trials outside of a master protocol framework. We have made the data described in this paper available via an interactive web site (<https://cancertrials.io/>) that we will continue to expand with new imputed data and analysis. We also welcome the submission of primary data.

Ongoing data collection efforts relevant to clinical trials include ~~the US National Cancer Institute's (NCI) Project DataSphere~~ Data Sphere⁶⁸, the ~~NCI's~~ NCI's National Clinical Trials Network (NCTN) and Community Oncology Research Program (NCORP) Data Archive, and The Yale University Open Data Access (YODA) Project⁶⁹. Unfortunately these projects have substantial limitations with respect to the type of analysis presented here: (i) most IPD are greater than six years old and do not cover many of the drugs of greatest current interest, including ICIs; (ii) most public data derives from control, not experimental treatment arms; (iii) much of the data involves summary statistics, not IPD, and requests for underlying data can be strictly limited; (iv) if access to IPD is granted, they are often available online for inspection but are not downloadable for computational analysis. A substantial unmet need therefore exists for primary data from clinical trials to be made available for reuse. One approach is to amend the requirements for data deposition on ClinicalTrials.gov (per U.S. Public Law 110-85) to include IPD.

METHODS

Individual participant data imputation and curation

The original data set consisted of 152153 unique trials in breast, colorectal, lung, and prostate cancer in the metastatic and non-metastatic settings from 2014-2016²¹. Trials were removed from the original data set if there were any inconsistencies in the imputed patient data as compared to its

associated clinical trial (e.g.: differing numbers of patients from the publication at-risk table and imputed data). The quality of the data imputation was confirmed quantitatively, by calculating the hazard ratio for imputed data and comparing it to the corresponding trial's reported hazard ratio, and qualitatively, by overlaying the Kaplan-Meier curve generated from the imputed data on top of the published curve. Trials with a hazard ratio difference greater than 0.1, or with perceptible visual differences, were removed from the final data set and not analyzed further (**Supplementary Data ~~File S1~~:1**). Trials enrolling patients with malignant pleural mesothelioma (e.g. MAPS, NCT00651456) were classified as “lung” trials.

Parametric fitting of patient survival data

The imputed event times ~~for the imputed patients~~, either death for overall survival distributions or surrogate events in the case of event-free survival distributions, were compared to the event times simulated under each parametric distribution. The likelihood of a specific parametric form to fit patient data was computed by maximum likelihood estimation. Specifically, the relative likelihood of a patient event taking place at a particular point in time was calculated under that parametric distribution's probability density function. The likelihood of a censoring event taking place was calculated by integrating the probability density function (the cumulative density function), and computing the likelihood of a patient event taking place in the trial after the censoring time (1- the cumulative probability at up to that time ~~under the cumulative density function~~). This procedure was repeated for all patient events in an arm of a clinical trial, and the overall likelihood of a fit was calculated by multiplying all relative likelihoods.

Computing R^2 explained by the Weibull fit

For imputed patient events in a clinical trial arm, the event times (deaths or surrogate events) and corresponding percent survival (OS or event-free survival) were computed. Weibull parametric fitting was used to obtain the best-fit α and β values corresponding to the imputed patient data. The differences between the survival distribution under a best-fit Weibull model and the imputed data were analyzed through a Weibull plot⁷⁰. In this approach, the event times and corresponding survival are normalized such that if the data follow a Weibull distribution, the points will be linear. The event times were normalized through the transformation: $\ln t/\beta$, while survival was normalized by: $\ln (-\ln S (t))/ \alpha$. Coefficient of determination (R^2) values were calculated to ~~determine~~assess the goodness of Weibull fitting for all trial arms in the data set.

Computing Weibull fits to trials of immune checkpoint inhibitors

Trials of immune checkpoint inhibitors were selected from the data set (five in total). Each trial's OS and PFS IPD were fit to a 1) single Weibull distribution 2) a mixture distribution made of two Weibull distributions. ~~The quality of all fits was assessed by computing the variance in event time explained by the fit and the Akaike information criterion (AIC).~~ An additional set of simulations was performed to account for the periodicity of radiological scans in detecting progression events, ~~and~~. The quality of fit ~~was~~for these simulations is not readily interpretable through use of a Weibull plot, and was instead quantified by the coefficient of determination (R^2). ~~between observed and fitted PFS.~~

Assessing the relationship between trial scan timetimes and PFS dropdrops

Trials of immune checkpoint inhibitors in oncology were obtained through a PubMed search of the terms "neoplasms" or "cancer" and "Clinical Trial, Phase III" along with therapies of interest ("ipilimumab" or "pembrolizumab" or "nivolumab"). The search was filtered to yield 25 trials with PFS

data and a reported scan time, in addition to the five trials in the original data set, for a total of 30 trials for subsequent analysis. PFS curves were extracted from each of the trials and images were analyzed using DigitizeIt software²² (Braunschweig, Germany) to estimate the timing of the PFS drop in each survival curve. The trial scan time interval was obtained from each publication's methods, blinded ~~from~~ the image associated with each trial. All extracted values can be found in **Supplementary Data File S3-3**.

Simulating differences in trial success based on α and β values

Control and experimental arms of clinical trials were simulated 1000 times by drawing 100 patient events from Weibull distributions with differing α and β values. α values in the experimental arm ranged from 0.5 to 4.5, β values from 0.4 to 1.6, and control arm parameter values were kept constant (1.5 and 1 respectively). Figure 7 shows results in the region of interest, from experimental arm $\alpha=0.5$ to 2.5. Events were censored at either early time points (corresponding to a ~60% event rate, a time equal to the control arm β value) or later time points (corresponding to a ~95% event rate, a time equal to four times the control arm β value). Trial success was calculated for each simulation by using a Cox regression at a significance level of $p=0.05$, ~~in accordance with standard statistical methods used in clinical trials~~. Significance was calculated using a Wald test.

Calculating the precision and accuracy of ~~parametric and nonparametric~~ survival estimates across sample sizes through subsampling

IPD for each trial arm in the data set was extracted. 213 OS and 273 event-free survival trial arms were used for further analysis; these trial arms had at least 100 patients (the maximum number of patient subsampling events used in this experiment) and at least one event (i.e.: death, progression)

taking place before 12 months. For each trial arm, 20-100 patient events (with a step size of 10 events) were subsampled from the imputed IPD. At least 3 non-censoring events were selected during each sampling simulation. This procedure was repeated ten times per sample size and trial. Parametric and nonparametric 95% confidence intervals for 12-month survival were computed for every sampling simulation.

Accuracy and precision plots were constructed for the subset of simulated trial arms returning numerical nonparametric confidence intervals (125 OS trial arms and 99 event-free survival trial arms). Note that nonparametric estimates did not return a numerical confidence interval for 41% of OS trial arms and 64% of event-free survival trial arms, while Weibull fitting made it possible to calculate 12-month confidence intervals for every trial arm in every simulation.

Quantification and statistical analysis tools

Analysis was performed using Wolfram Mathematica Version 12.1.0.0. Details of the statistical analysis performed, exact values of n and what they represent, definitions of the summary statistics used, definitions of significance, and trial inclusion and exclusion criteria can be found in the Method Details, Figure captions, and Results sections of the manuscript. Compute-intensive analyses (e.g. sample size simulations) were conducted on the O2 High Performance Compute Cluster, supported by the Research Computing Group, at Harvard Medical School.

DATA AVAILABILITY

All data generated or analyzed during this study are included in this published article (and its supplementary information files).

CODE AVAILABILITY

All code used in this study is included in Supplementary Data ~~File S2. Each piece of code is provided in a folder containing a Mathematica Notebook (.nb), all data required by the code, and the corresponding code output. With source data kept within the same folder as the code, the Mathematica Notebook can be executed in Wolfram Mathematica version 11 by selecting “Evaluate Notebook” from the “Evaluation” menu.~~

2. Each piece of code is provided in a folder containing a Mathematica Notebook (.nb), all data required by the code, and the corresponding code output. With source data kept within the same folder as the code, the Mathematica Notebook can be executed in Wolfram Mathematica by selecting “Evaluate Notebook” from the “Evaluation” menu. Sample R code illustrates the parametric fitting and confidence interval construction procedures.

REFERENCES

1. Lindner, M. D. Clinical attrition due to biased preclinical assessments of potential efficacy. *Pharmacology & Therapeutics* **115**, 148–175 (2007).
2. Zhu, A. Z. Quantitative translational modeling to facilitate preclinical to clinical efficacy & toxicity translation in oncology. *Future Science OA* **4**, FSO306 (2018).
3. Lin, A. *et al.* Off-target toxicity is a common mechanism of action of cancer drugs undergoing clinical trials. *Sci. Transl. Med.* **11**, eaaw8412 (2019).
4. Haidich, A. B. Meta-analysis in medical research. *Hippokratia* **14**, 29–37 (2010).
5. Whitehead, A. *Meta-analysis of controlled clinical trials.* (John Wiley & Sons, 2002).
6. Weimer, K. & Enck, P. Traditional and innovative experimental and clinical trial designs and their advantages and pitfalls. *Handb Exp Pharmacol* **225**, 237–272 (2014).
7. Stewart, L. A. & Tierney, J. F. To IPD or not to IPD?: Advantages and Disadvantages of Systematic Reviews Using Individual Patient Data. *Eval Health Prof* **25**, 76–97 (2002).

8. Riley, R. D., Lambert, P. C. & Abo-Zaid, G. Meta-analysis of individual participant data: rationale, conduct, and reporting. *BMJ* **340**, c221 (2010).
9. Kaplan, E. L. & Meier, P. Nonparametric Estimation from Incomplete Observations. **53**, (1958).
10. Wan, X., Peng, L. & Li, Y. A Review and Comparison of Methods for Recreating Individual Patient Data from Published Kaplan-Meier Survival Curves for Economic Evaluations: A Simulation Study. *PLoS ONE* **10**, e0121353 (2015).
11. Taichman, D. B. *et al.* Data Sharing Statements for Clinical Trials — A Requirement of the International Committee of Medical Journal Editors. *N Engl J Med* **376**, 2277–2279 (2017).
12. Danchev, V., Min, Y., Borghi, J., Baiocchi, M. & Ioannidis, J. P. A. Evaluation of Data Sharing After Implementation of the International Committee of Medical Journal Editors Data Sharing Statement Requirement. *JAMA Netw Open* **4**, e2033972 (2021).
13. Alexander, B. M., Schoenfeld, J. D. & Trippa, L. Hazards of Hazard Ratios - Deviations from Model Assumptions in Immunotherapy. *N. Engl. J. Med.* **378**, 1158–1159 (2018).
14. Guyot, P., Ades, A., Ouwens, M. J. & Welton, N. J. Enhanced secondary analysis of survival data: reconstructing the data from published Kaplan-Meier survival curves. *BMC Med Res Methodol* **12**, 9 (2012).
15. Rahman, R. *et al.* Deviation from the Proportional Hazards Assumption in Randomized Phase 3 Clinical Trials in Oncology: Prevalence, Associated Factors, and Implications. *Clin Cancer Res* **25**, 6339–6345 (2019).
16. Committee on Strategies for Responsible Sharing of Clinical Trial Data, Board on Health Sciences Policy, & Institute of Medicine. *Discussion Framework for Clinical Trial Data Sharing: Guiding Principles, Elements, and Activities*. (National Academies Press (US), 2014).
17. Boag, J. W. Maximum Likelihood Estimates of the Proportion of Patients Cured by Cancer Therapy. *Journal of the Royal Statistical Society. Series B (Methodological)* **11**, 15–53 (1949).
18. Massett, H. A. *et al.* Challenges Facing Early Phase Trials Sponsored by the National Cancer Institute: An Analysis of Corrective Action Plans to Improve Accrual. *Clin Cancer Res* **22**, 5408–5416 (2016).
19. Ferrara, R. *et al.* Do immune checkpoint inhibitors need new studies methodology? *Journal of Thoracic Disease* **1**, (2018).
20. Hoyle, M. W. & Henley, W. Improved curve fits to summary survival data: application to economic evaluation of health technologies. *BMC Med Res Methodol* **11**, 139 (2011).
21. Fell, G. *et al.* KMDATA: a curated database of reconstructed individual patient-level data from 153 oncology clinical trials. *Database* **2021**, baab037 (2021).

22. Rakap, S., Rakap, S., Evran, D. & Cig, O. Comparative evaluation of the reliability and validity of three data extraction programs: UnGraph, GraphClick, and DigitizeIt. *Computers in Human Behavior* **55**, 159–166 (2016).
23. Caldwell, D. Decision Modelling for Health Economic Evaluation. A Briggs, M Sculpher, K Claxton. *International Journal of Epidemiology* **36**, 476–477 (2007).
24. Collett, D. *Modelling survival data in medical research*. (Chapman & Hall/CRC, 2003).
25. Klein, J. P. & Moeschberger, M. L. *Survival analysis: techniques for censored and truncated data*. (Springer, 2003).
26. Lawless, J. F. *Statistical models and methods for lifetime data*. (Wiley-Interscience, 2003).
27. Kalbfleisch, J. D. & Prentice, R. L. *The statistical analysis of failure time data*. (J. Wiley, 2002).
28. Cox, D. R. & Oakes, D. *Analysis of Survival Data*. (Chapman and Hall, 1988).
29. Nelson, W. B. *Applied Life Data Analysis*. (John Wiley & Sons, 2003).
30. Kleinbaum, D. G. & Klein, M. *Survival analysis: a self-learning text*. (Springer, 2012).
31. Matsushita, S. *et al.* Lifetime data analysis of disease and aging by the weibull probability distribution. *Journal of Clinical Epidemiology* **45**, 1165–1175 (1992).
32. Cox, D. R. Regression Models and Life-Tables. *Journal of the Royal Statistical Society. Series B (Methodological)* **34**, 187–220 (1972).
33. Peeters, M. *et al.* Final results from a randomized phase 3 study of FOLFIRI {+/-} panitumumab for second-line treatment of metastatic colorectal cancer. *Ann Oncol* **25**, 107–116 (2014).
34. Glynne-Jones, R. *et al.* Chronicle: results of a randomised phase III trial in locally advanced rectal cancer after neoadjuvant chemoradiation randomising postoperative adjuvant capecitabine plus oxaliplatin (XELOX) versus control. *Ann Oncol* **25**, 1356–1362 (2014).
35. Hagman, H. *et al.* A randomized study of KRAS-guided maintenance therapy with bevacizumab, erlotinib or metronomic capecitabine after first-line induction treatment of metastatic colorectal cancer: the Nordic ACT2 trial. *Ann Oncol* **27**, 140–147 (2016).
36. Yu, B., Tiwari, R. C., Cronin, K. A. & Feuer, E. J. Cure fraction estimation from the mixture cure models for grouped survival data. *Statist. Med.* **23**, 1733–1747 (2004).
37. Schmittlutz, K. & Marks, R. Current treatment options for aggressive non-Hodgkin lymphoma in elderly and frail patients: practical considerations for the hematologist. *Therapeutic Advances in Hematology* **12**, 2040620721996484 (2021).

38. Stewart, D. J. *et al.* Abstract 1774: Progression-free survival curves suggest a dichotomous determinant of PD-L1 inhibitor efficacy. in *Clinical Research (Excluding Clinical Trials) 1774–1774* (American Association for Cancer Research, 2017). doi:10.1158/1538-7445.AM2017-1774.
39. Gibson, E. *et al.* Modelling the Survival Outcomes of Immuno-Oncology Drugs in Economic Evaluations: A Systematic Approach to Data Analysis and Extrapolation. *Pharmacoeconomics* **35**, 1257–1270 (2017).
40. Kok, P.-S. *et al.* Validation of Progression-Free Survival Rate at 6 Months and Objective Response for Estimating Overall Survival in Immune Checkpoint Inhibitor Trials: A Systematic Review and Meta-analysis. *JAMA Netw Open* **3**, e2011809 (2020).
41. Ribas, A. *et al.* Combined BRAF and MEK inhibition with PD-1 blockade immunotherapy in BRAF-mutant melanoma. *Nat Med* **25**, 936–940 (2019).
42. Ascierto, P. A. *et al.* Dabrafenib, trametinib and pembrolizumab or placebo in BRAF-mutant melanoma. *Nat Med* **25**, 941–946 (2019).
43. Allemani, C. *et al.* Global surveillance of cancer survival 1995–2009: analysis of individual data for 25 676 887 patients from 279 population-based registries in 67 countries (CONCORD-2). *The Lancet* **385**, 977–1010 (2015).
44. National Cancer Institute, SEER. SEER Incidence Database - SEER Data & Software.
<https://seer.cancer.gov/data/index.html>.
45. Palmer, A. C., Plana, D. & Sorger, P. K. Comparing the Efficacy of Cancer Therapies between Subgroups in Basket Trials. *Cell Systems* **11**, 449-460.e2 (2020).
46. Park, J. J. H., Hsu, G., Siden, E. G., Thorlund, K. & Mills, E. J. An overview of precision oncology basket and umbrella trials for clinicians. *CA A Cancer J Clin* **70**, 125–137 (2020).
47. Rulli, E. *et al.* Assessment of proportional hazard assumption in aggregate data: a systematic review on statistical methodology in clinical trials using time-to-event endpoint. *Br J Cancer* **119**, 1456–1463 (2018).
48. Grambsch, P. M. & Therneau, T. M. Proportional Hazards Tests and Diagnostics Based on Weighted Residuals. *Biometrika* **81**, 515–526 (1994).
49. Borghaei, H. *et al.* Nivolumab versus Docetaxel in Advanced Nonsquamous Non-Small-Cell Lung Cancer. *N Engl J Med* **373**, 1627–1639 (2015).
50. Kwon, E. D. *et al.* Ipilimumab versus placebo after radiotherapy in patients with metastatic castration-resistant prostate cancer that had progressed after docetaxel chemotherapy (CA184-043): a multicentre, randomised, double-blind, phase 3 trial. *Lancet Oncol* **15**, 700–712 (2014).

51. Ferrucci, P. F. *et al.* KEYNOTE-022 part 3: a randomized, double-blind, phase 2 study of pembrolizumab, dabrafenib, and trametinib in *BRAF* -mutant melanoma. *J Immunother Cancer* **8**, e001806 (2020).
52. Chen, T.-T. Statistical issues and challenges in immuno-oncology. *j. immunotherapy cancer* **1**, 18 (2013).
53. Mick, R. & Chen, T.-T. Statistical Challenges in the Design of Late-Stage Cancer Immunotherapy Studies. *Cancer Immunol Res* **3**, 1292–1298 (2015).
54. Ouwens, M. J. N. M. *et al.* Estimating Lifetime Benefits Associated with Immuno-Oncology Therapies: Challenges and Approaches for Overall Survival Extrapolations. *Pharmacoeconomics* **37**, 1129–1138 (2019).
55. Kumar, S. K. *et al.* Venetoclax or placebo in combination with bortezomib and dexamethasone in patients with relapsed or refractory multiple myeloma (BELLINI): a randomised, double-blind, multicentre, phase 3 trial. *The Lancet Oncology* **21**, 1630–1642 (2020).
56. Wu, J. Power and Sample Size for Randomized Phase III Survival Trials Under the Weibull Model. *Journal of Biopharmaceutical Statistics* **25**, 16–28 (2015).
57. Heo, M., Faith, M. S. & Allison, D. B. Power and sample size for survival analysis under the Weibull distribution when the whole lifespan is of interest. *Mechanisms of Ageing and Development* **102**, 45–53 (1998).
58. Jiang, Z., Wang, L., Li, C., Xia, J. & Jia, H. A Practical Simulation Method to Calculate Sample Size of Group Sequential Trials for Time-to-Event Data under Exponential and Weibull Distribution. *PLoS ONE* **7**, e44013 (2012).
59. Lu, Q., Tse, S.-K., Chow, S.-C. & Lin, M. Analysis of time-to-event data with nonuniform patient entry and loss to follow-up under a two-stage seamless adaptive design with weibull distribution. *J Biopharm Stat* **22**, 773–784 (2012).
60. Liu, S. & Yuan, Y. Bayesian optimal interval designs for phase I clinical trials. *J. R. Stat. Soc. C* **64**, 507–523 (2015).
61. Wong, C. H., Siah, K. W. & Lo, A. W. Estimation of clinical trial success rates and related parameters. *Biostatistics* **20**, 273–286 (2019).
62. Kolata, G. A Cancer Conundrum: Too Many Drug Trials, Too Few Patients. *The New York Times* (2017).
63. Fleming, T. R. & Harrington, D. P. *Counting processes and survival analysis*. (Wiley-Interscience, 2005).
64. Yang, S. & Prentice, R. Improved Logrank-Type Tests for Survival Data Using Adaptive Weights. *Biometrics* **66**, 30–38 (2010).
65. Royston, P. & Parmar, M. K. Restricted mean survival time: an alternative to the hazard ratio for the design and analysis of randomized trials with a time-to-event outcome. *BMC Med Res Methodol* **13**, 152 (2013).
66. Uno, H. *et al.* Moving Beyond the Hazard Ratio in Quantifying the Between-Group Difference in Survival Analysis. *JCO* **32**, 2380–2385 (2014).

67. Arfè, A., Alexander, B. & Trippa, L. Optimality of testing procedures for survival data in the nonproportional hazards setting. *Biometrics* biom.13315 (2020) doi:10.1111/biom.13315.
68. Hede, K. Project Data Sphere to Make Cancer Clinical Trial Data Publicly Available. *JNCI Journal of the National Cancer Institute* **105**, 1159–1160 (2013).
69. Ross, J. S. *et al.* Overview and experience of the YODA Project with clinical trial data sharing after 5 years. *Sci Data* **5**, 180268 (2018).
70. Nelson, W. *Applied life data analysis*. (Wiley-Interscience, 2004).

ACKNOWLEDGMENTS

We thank Lorenzo Trippa, Andrea Arfe, Giovanni Parmigiani, Charles Perou, John Higgins, Michael Kosorok, Nicholas Clark, Clemens Hug, and Claire Lazar Reich for their helpful comments on this project. We thank Jeremy Muhlich and Zev Ross for their assistance in implementing the <https://cancertrials.io/> site. We thank William Sorger for his assistance in recording the trial metadata. We are grateful to all of the patients and investigators who participated in the clinical trials analyzed in this work. This project was supported by NIH grants P50-GM107618 and U54-CA225088 (to P.K.S). D.P. is supported by NIGMS grant T32-GM007753 and F30-CA260780.

AUTHOR INFORMATION

Affiliations

Laboratory of Systems Pharmacology and the Department of Systems Biology, Harvard Medical School, Boston, Massachusetts, USA

Deborah Plana & Peter K. Sorger

Harvard-MIT Division of Health Sciences and Technology, Harvard Medical School and MIT, Cambridge, Massachusetts, USA

Deborah Plana

Dana-Farber Cancer Institute, Boston, Massachusetts, USA

~~Geoffrey Fell & Brian M. Alexander~~

~~Foundation Medicine Inc., Cambridge, Massachusetts, USA~~

~~Brian M. Alexander~~

~~Department of Pharmacology, Computational Medicine Program, Lineberger Comprehensive Cancer Center, University of North Carolina at Chapel Hill, Chapel Hill, North Carolina, USA~~

~~Adam C. Palmer~~

Author Contributions

Generating individual participant data set: G.F., B.M.A.

Metadata curation; data analysis: D.P. and A.C.P.

Writing: D.P., A.C.P., and P.K.S.

Manuscript review and editing: D.P., G.F., B.M.A., A.C.P., P.K.S.

Supervision: B.M.A., A.C.P., P.K.S.

Funding: B.M.A. and P.K.S.

Corresponding Author

~~Correspondence to Adam C. Palmer and Peter K. Sorger: palmer@unc.edu;~~

~~peter_sorger@hms.harvard.edu (cc: sorger_admin@hms.harvard.edu).~~

ETHICS DECLARATIONS

Competing Interests

P.K. Sorger is a member of the SAB or Board of Directors of Applied Biomath, Glencoe Software,

RareCyte Inc and ~~NanoString and~~ has equity in ~~the first three of~~ these companies; he is a member of the SAB of NanoString Inc and a consultant for Merck and Montai Health. In the last five years the Sorger

lab has received research funding from Novartis and Merck. Sorger declares that none of these

relationships are directly or indirectly related to the content of this manuscript. B.M. Alexander is an

employee of Foundation Medicine. No potential conflicts of interest were disclosed by the other authors.

SUPPLEMENTARY INFORMATION

Supplementary Data File S1. Clinical trial metadata and IPD. Trial metadata file includes: trial name, author, registration number, journal, publication date, cancer type, cancer metastatic status, whether a significant difference was found between the trial experimental and control arm, treatment name, treatment type, and number of patients enrolled in the trial. Comparisons between the imputed trials' hazard ratios and the original trial hazard ratios are included to assess imputation quality (procedure described in Methods). IPD is provided as 262 .csv files. Each .csv file contains IPD from a different figure from a published clinical trial. Description of all variables included in metadata and .csv files can be found in "README.txt".

Supplementary Data File S2. Analysis code. Each piece of code is provided in a folder containing a Mathematica Notebook (.nb), all data required by the code, and the corresponding code output. With source data kept within the same folder as the code, the Mathematica Notebook can be executed in Wolfram Mathematica version 11 by selecting "Evaluate Notebook" from the "Evaluation" menu.

Supplementary Data File S3. Weibull fitting of ICI trial arms. The first tab contains percent variance explained and AIC for one and two distribution fits of ICI trial arms. The second tab contains the timing of ICI trial PFS drops and the corresponding trial scan times.

FIGURE TITLES AND LEGENDS

A. Reconstruct event tables from published Kaplan-Meier estimators and at-risk tables

B. Calculate likelihood of observing patient data given a set of Weibull parameters

$$\text{Relative likelihood of survival function with } \alpha=1.2, \beta=13 = \prod P_{\alpha=1.2, \beta=13}(t_{\text{Death}}) \times \prod P_{\alpha=1.2, \beta=13}(t_{\text{Censoring}})$$

C. Compute likelihood for range of α and β

D. Fit with maximum likelihood

Fig 1: Procedure for parameterizing survival curves starting with published figures. (A) Kaplan-Meier survival curve and at-risk table obtained from clinical trial publication. Individual patient data (IPD) were imputed from digitized survival curves and at-risk tables as previously described (see Methods). (B) Each set of parameters corresponds to a different probability density function (PDF) and survival function (which corresponds to 1- the cumulative density function (CDF)). The likelihood of observing actual data is then computed. (C) Likelihood calculation is repeated for all possible parameter values. (D) The most likely (best) fit is obtained by finding the parameter values with the maximum likelihood.

Fig. 2: Representative fits of Weibull distributions to overall survival trial data. (A) Weibull fits to data for plots of the Kaplan-Meier estimator falling in the top 25th percentile quality of all fits (NCT00973609, NCT01212991) (B) at the 50th percentile (NCT00003140/NCT00754845, NCT01607957) and (C) in the bottom 25th percentile (NCT00427713, NCT01229813). (D) Weibull plot for patient events in all OS trials (for 237 trial arms from 116 publication figures). This is a transformation of survival data that is linear if the data follow a Weibull distribution. (E) Goodness of fit as the coefficient of determination (R^2) explained by fitted Weibull functions for all trial arms reporting OS data.

Supplementary Fig. S1. Representative fits of Weibull distributions to event-free survival data.

Weibull fits to data for plots of the Kaplan-Meier estimator falling in the top 25th percentile quality for all fits (excellent fits; NCT00294996, NCT00053898), at the 50th percentile (good fits; NCT00402519, NCT01001377), and in the bottom 25th percentile (poor fits; NCT00427713, NCT number not reported). Data derived from trials reporting event-free survival data, primarily PFS (301 survival curves from 146 figures).

Supplementary Fig. S2. Improvements in fit of Weibull distributions to overall survival data using three-parameter models. Two-parameter fits for the trial arms in Figure 2 representative of the bottom quartile of all Weibull fits (NCT00427713, NCT01229813) and two-parameter fits with an additional cure rate parameter.

A. Biomarker positive (mutant *KRAS*)

B. Biomarker negative (wild-type *KRAS*)

Supplementary

FIGURES

a Reconstruct event tables from published Kaplan-Meier estimators and at-risk tables

b Calculate likelihood of observing patient data given a set of Weibull parameters

$$\text{Relative likelihood of survival function with } \alpha=1.2, \beta=13 = \prod P_{\alpha=1.2, \beta=13}(t_{\text{Death}}) \times \prod P_{\alpha=1.2, \beta=13}(t_{\text{Censoring}})$$

c Compute likelihood for range of α and β

d Fit with maximum likelihood

Fig 1: Procedure for parameterizing survival curves starting with published figures. a

Kaplan-Meier survival curve and at-risk table obtained from clinical trial publication. Individual participant data (IPD) were imputed from digitized survival curves and at-risk tables as previously described (see Methods). **b** Each set of parameters corresponds to a different probability density function (PDF) and survival function (which corresponds to 1- the cumulative density function (CDF)). The likelihood of observing actual data is then computed. **c** Likelihood calculation is repeated for a set of possible parameter values. **d** The most likely (best) fit is obtained by finding the parameter values with the maximum likelihood.

Fig. 2: Representative fits of Weibull distributions to overall survival trial data. **a** Weibull fits to data for plots of the Kaplan-Meier estimator falling in the top 25th percentile quality of all fits (NCT00973609, NCT00982111) **b** between the first and third quartile (NCT01377376, NCT00601900) and **c** in the lower quartile (NCT00427713, NCT01229813). **d** Weibull plot for patient events in all OS trials (for 237 trial arms from 116 publication figures). This is a transformation of survival data that is linear if the data follow a Weibull distribution. **e** Goodness of fit as the coefficient of determination (R^2) explained by fitted Weibull functions for all trial arms reporting OS data. Boxplot center line indicates median; box limits indicate the upper and lower quartile values; whiskers span the full dataset.

a Progression-free survival

b Overall survival

c Weibull fit where progression is observed at scan times

Fig. S3. Fit of Weibull models to overall survival and progression free survival data from biomarker stratified trials. Data obtained from the 20050181 trial (NCT number not reported).

A. Progression-free survival

B. Overall survival

— Trial data — One Weibull distribution — Two Weibull distributions

C. Weibull fit where progression is observed at scan times

— Trial data — One Weibull distribution — Weibull distribution with scan time

Fig. 3: Fit of Weibull models to overall survival and progression-free survival data for trials of immune checkpoint inhibitors. (A) Progression-free survival PFS distributions and (B) overall survival OS distributions for individual arms of two trials of immune checkpoint inhibitors (NCT01673867, NCT01057810) with fit to either one or two Weibull distributions. PFS data are best described by using a mixture model of two Weibull distributions, each with two parameters. OS distributions are described equivalently well by a one or two-distribution fit. (C) PFS simulations that account for the periodicity of radiological scans to detect progression improve the quality of one-distribution Weibull fits, as quantified by the coefficient of determination (R^2).

Supplementary

~~Fig-S4. Additional fits to overall survival and progression-free survival data for trials of immune checkpoint inhibitors. Progression-free survival and overall survival distributions for three trials of immune checkpoint inhibitors with one and two distribution fits using Weibull functions, as well as simulations that account for the periodicity of radiological scans to detect progression (NCT01642004, NCT02142738, NCT00861614).~~

Fig. 4: Impact of parametric statistics on precision and accuracy of overall survival confidence interval estimates. Comparison of nonparametric and parametric (Weibull distribution) methods to compute 12-month ~~overall survival~~OS confidence intervals for trials with small cohorts (20 to 100 patients) produced by randomly subsampling patient events from 125 actual trial arms. Note that nonparametric estimates did not return ~~a numerical~~an informative confidence interval for 41% of OS curves; (out of 213 OS curves with at least 100 patients), while Weibull fitting made it possible to calculate 12-month confidence intervals for every survival curve in every simulation. Precision is defined as the width of the confidence interval in percent survival. Accuracy is defined as the absolute difference between the 12-month survival estimated from small cohorts (for each of 10 simulations per sample size and trial), and the value computed from all patients in a given Phase ~~III~~3 trial. Lines denote mean values and shaded regions are 95% confidence intervals.

Fig.

Supplementary Fig. S5. Impact of parametric statistics on precision and accuracy of event-free survival estimates. (A) Comparison of nonparametric and parametric (Weibull distribution) methods to compute 12-month event-free survival confidence intervals for trials with small cohorts (20 to 100 patients) produced by randomly subsampling patient events from 99 event-free survival trial arms. Note that nonparametric estimates did not return a numerical confidence interval for 64% of event-free survival curves, while Weibull fitting made it possible to calculate 12-month confidence intervals for every survival curve in every simulation. Precision is defined as the width of the confidence interval in percent survival. Accuracy is defined as the absolute difference between the 12-month survival estimated from small cohorts (for each of 10 simulations per sample size and trial), and the value computed from all patients in a given phase III trial. Shaded regions are 95% confidence intervals. **(B)** Median PFS confidence intervals calculated with parametric and nonparametric methods on phase I clinical trial data, and the corresponding phase II study results (MK-3475-022/KEYNOTE-022; NCT02130466).

Fig. 5: Best fit Weibull parameter values for trials reporting overall survival data. Weibull fits for trials reporting overall survival data. **(A)** 5: Best-fit Weibull parameter values for trials reporting overall survival data. Weibull fits for trials reporting OS data, encompassing 237 survival curves from 116 trial figures. **a** Survival distributions categorized by cancer type and metastatic status (defined as trials that included patients with distant metastases). Representative survival functions and fits for trials across a variety of cancer types including **(B)** h metastatic colorectal cancer (NCT01584830) **(C)** c non-metastatic lung cancer (NCT number not reported) **(D)** d metastatic prostate cancer (NCT00110162) and **(E)** e non-metastatic breast cancer (NCT number not reported).

Supplementary Fig. S6. Best fit Weibull parameter values for trials reporting event free survival data. Weibull fits for event free survival curves labeled by metastatic status and cancer type (encompassing 301 survival curves from 146 trial figures).

A. Weibull Parameters for OS Trials

B. ATTENTION C. RECOURSE D. CA184-043 E. CheckMate 057

Fig. 6: Parameter values for Weibull fits to overall survival data scored by trial outcome. (A)a Differences in Weibull α and β parameters for experimental and control arm OS data (drawn from 116 trial figures). For α , the value for the control arm was subtracted from the value for the experimental arm. Differences in β were computed by determining the percent change in β in the experimental arm with respect to the control arm (positive values indicate larger β in the experimental arm). Asterisks denote trials that tested immune checkpoint inhibitors. Success in all cases was judged based on the original report and most often corresponded to $HR < 1$ at 95% confidence by based on Cox proportional hazards regression. Hazard ratio and Weibull ratio of cumulative hazards of four clinical trials in the data set: **(B)b** NCT01377376 (tivantinib plus erlotinib vs. erlotinib), **(C)c** NCT01607957 (TAS-102 vs. placebo), **(D)d** NCT00861614 (ipilimumab vs. placebo), **(E)e** NCT01673867 (nivolumab vs. docetaxel). The uncertainty in estimating the ratio of cumulative hazards typically falls with time, narrowing the

95% confidence interval depicted in blue.), e NCT00861614 (ipilimumab vs. placebo after radiotherapy).

Supplementary Fig. S7. Parameter values for Weibull fits to event-free survival data scored by outcome of the trial. For α , the value in the control arm was subtracted from the value for the experimental arm. Differences in β were computed by determining the percent change in β value in the experimental arm with respect to the control arm (positive values indicate larger β in the experimental arm). Success in all cases was judged based on the original report and most often corresponded to HR<1 at 95% confidence by Cox proportional hazards regression.

A.

B.

C.

Fig. 7: Effect of trial duration on success when proportional hazards is violated. (A) One of many simulated trials having a range of Weibull α and β parameters similar to those observed in actual trials reporting OS data; in this trial $\Delta\alpha = -0.5$ and the ratio of β for experimental and control arms was 1.2. The labels t_A and t_B denote times corresponding to approximately 60% or 95% of all trial events (for real OS trial arms in this article, in metastatic cancers, these event rates correspond to median times of $t_A = 1622$ months and $t_B = 5150$ months). **(B)** Percent of simulated successful trials at time t_A (left panel) or t_B (right panel). The simulated trial depicted in panel A is denoted by an asterisk. “Success” was scored as $HR < 1$ at a 95% confidence level using the Cox proportional hazards regression; this metric was used despite the violation of proportional hazards because it is the accepted approach for assessing efficacy in pivotal trials (see text and Methods for details).

Reviewers' Comments:

Reviewer #1:

Remarks to the Author:

I would like to appreciate the author's thorough response to address my comments. No additional comments

Reviewer #2:

Remarks to the Author:

Thank you for addressing my concerns. I do not have any further comments.